# Soil carbon response to land-use change: Evaluation of a global vegetation model using observational meta-analyses

Sylvia S. Nyawira[1,2], Julia E. M. S. Nabel[1], Axel Don[3], Victor Brovkin[1], and Julia Pongratz[1]

[1]Max Planck Institute for Meteorology, Hamburg, Germany
[2]International Max Planck Research School on Earth System Modelling, Hamburg, Germany
[3]Thünen Institute of Climate-Smart Agriculture, Braunschweig, Germany

*Correspondence to:* Sylvia Sarah Nyawira (sylvia.nyawira@mpimet.mpg.de)

**Abstract.** Global model estimates of soil carbon changes from past land-use changes remain uncertain. We develop an approach for evaluating dynamic global vegetation models (DGVMs) against existing observational meta-analyses on soil carbon changes following land-use change. Using the DGVM JSBACH, we perform idealized simulations where the entire globe is covered by one vegetation type, which then undergoes a land-use change to another vegetation type. We select the grid cells that represent the climatic conditions of the meta-analyses and compare the mean simulated soil carbon changes to the meta-analyses. Our simulated results show model agreement with the observational data on the direction of changes in soil carbon for some land-use changes, although the model simulated generally smaller magnitude of changes. The conversion of crop to forest resulted in soil carbon gain of 10% compared to a gain of 42% in the data, whereas the forest to crop resulted in a simulated loss of -15% compared to -40%. The model and the observational data disagreed for the conversion of crop to grasslands. The model estimated a small soil carbon loss (-4%), while observational data indicate a 38% gain in soil C for the same land-use change. These model deviations from the observations are substantially reduced by explicitly accounting for crop harvesting and neglecting burning in grasslands in the model. We conclude that our idealized simulation approach provides an appropriate framework for evaluating DGVMs against meta-analyses and that this evaluation helps to identify the causes of deviation of simulated soil carbon changes from the meta-analyses.

## 1   Introduction

Global model estimates of land-use-related soil carbon (soil C) changes rely on dynamic global vegetation models (DGVMs). To judge the reliability of DGVMs in simulating past and future changes of soil C, models have to be evaluated against observations. A range of meta-analyses on soil C changes following land-use change (LUC) has been published recently, aggregating local-scale measurements to spatial scales potentially applicable to DGVMs (e.g., Guo and Gifford, 2002). Here, we develop an approach for evaluating DGVMs against the observational data and apply the approach to evaluate the DGVM JSBACH.

A major driver of soil C changes in recent centuries has been LUC. For example, the replacement of natural vegetation with croplands usually leads to soil C loss while the reverse leads to a gain (Guo and Gifford, 2002). Unlike for vegetation, soil dynamics include slower processes ranging from decadal to centennial timescales; hence the carbon response to LUC lags

the changes in vegetation carbon. Soil C changes due to LUC are caused by changes in soil C inputs and outputs when one vegetation type is replaced by another. Changes in soil C inputs stem from differences in litter quality and quantity, while the changes in outputs stem from alteration of soil decomposition processes that govern stabilisation of carbon in soils. The response of soil C to LUC depends on the local conditions, such as soil type, mineralogy and texture (Lugo et al., 1986) and on climate influences, such as temperature and soil moisture or precipitation (Marín-Spiotta and Sharma, 2013). Also, management practices can influence the soil C response; for example, Poeplau and Don (2015) showed that planting cover crops during winter and tilling them into the soil as additional carbon input can significantly enhance soil C on croplands. Due to the slow response of soils to LUC, soil C changes from past LUCs continue to have a long-term effect on the global carbon budget (Pongratz et al., 2009).

Despite the dependence of the soil C response to local conditions of soils, climate and management practices, regional and global syntheses of published data can be useful to aggregate local-scale measurements on soil C changes and estimate mean responses to different LUCs using a meta-analyses approach. Over the recent past, several of these meta-analyses have been published (Post and Kwon, 2000; Guo and Gifford, 2002; Paul and Polglase, 2002; Murty et al., 2002; Laganiére et al., 2010; Poeplau et al., 2011; Don et al., 2011). An advantage of the meta-analyses is that they apply several quality checks to combine and aggregate the local-scale measurements. The meta-analyses provide estimates of the average magnitudes of relative changes and additionally the temporal response of soil C to LUC (Poeplau et al., 2011; Poeplau and Don, 2015). These analyses have also been used to understand the factors influencing the spatial and temporal variability of soil C changes following LUC. This has been done by correlating variables such as temperature, precipitation and clay content with the soil C changes (Don et al., 2011; Wei et al., 2014). However, the applicability of this observational data for global modeling has not been tested so far.

DGVMs are used to study the effects of LUC on soil C globally. They combine information on the past vegetation distribution, climate and LUC data and incorporate various processes to quantify global changes in terrestrial carbon stocks resulting from past LUCs (e.g. Pongratz et al., 2009; Stocker et al., 2011). In addition, by simulating climate and LUC scenarios following the representative concentration pathways, DGVMs are used to make future projections in terrestrial carbon stocks (e.g. Brovkin et al., 2013). However, global estimates of LUC carbon fluxes by different DGVMs show a large spread (Ciais et al., 2013). This spread has been attributed to several factors: the different climate used in driving the DGVMs (Anav et al., 2013), different modeling approaches of LUC (Houghton et al., 2012; Wilkenskjeld et al., 2014), inconsistent definition of land-use fluxes (Pongratz et al., 2014), parameterizations related to fluxes of land-use and land cover change (Brovkin et al., 2013; Goll et al., 2015) and land-management processes (Houghton et al., 2012). In a recent study, Tian et al. (2015) used the same model setup, with the same climate and LUC input data, to quantify global changes in soil C resulting from past LUCs across different DGVMs. They found that these changes differ widely across the models with some models showing almost no change and others showing a large decrease in soil C. Until now, soil C changes resulting from different LUCs in DGVMs have not been compared to observational data compiled by the different meta-analyses. This is because an approach for comparing these changes to the meta-analyses is still lacking and many of the meta-analyses have only become available relatively recently.

Our study aims at developing an approach that can be applied to any DGVM for evaluating the soil C changes following LUC against the meta-analyses. We test the applicability of the approach using the DGVM JSBACH and identify what the comparison reveals in terms of model processes. Further, we highlight the challenges involved in comparing simulated results to the meta-analyses and suggest what can be done to overcome these challenges. This is to our knowledge the first time that simulated soil C response to different LUCs in a DGVM are compared systematically to meta-analyses.

## 2 Methods

### 2.1 Meta-analyses

In this study, we use results from the meta-analyses by Poeplau et al. (2011) in the temperate regions and Don et al. (2011) for the tropical regions including 95 and 385 published studies, respectively. The published studies include sites from different countries in the tropics and temperate regions. The site studies were conducted using two main experimental designs: paired plots comparing soil C between two adjacent sites with different land use types, and time series where the soil C of a particular site was monitored overtime after LUC. The paired plot approach is used to construct chronosequences comprising of plots with different ages after LUC that use one of the plots as the reference site. The paired plot based approach goes along with a higher methodological uncertainty in the data due to differences in the inherent soil properties such as texture between the plots, which affect the response of soil C to LUC. In contrast, the time series observational data are without such uncertainties, but very few time series are available to investigate the response of soil C to LUC. In calculating the soil C changes across the different sites, the reference site was always assumed to be in equilibrium.

The meta-analyses defined the following criteria for including the site studies: (1) climate conditions, age of the current land use, and the relevant site characteristics such as soil type, texture and land-use history had to be provided, (2) studies on organic and wetland soils were not included and (3) for paired plots the sites had to be adjacent to each other to reduce uncertainties due to the spatial variability of soil properties unrelated to the LUC (Don et al., 2011; Poeplau et al., 2011). Any studies that did not match any of the criteria were excluded in the compilation. The soil bulk densities were used to calculate the soil organic carbon in $Mg/ha$. Mass correction was applied to account for changes in density with depth (Ellert and Bettany, 1995). In addition, Poeplau et al. (2011) used different variables, such as climate, time after LUC and the clay content, to derive carbon response functions (CRFs) describing the temporal response of soil C to LUC for the temperate regions. The response functions include general CRFs that account for only time after the LUC and specific CRFs that account for other site properties. Table 1 shows the LUCs represented in the two meta-analyses that are included in our study.

### 2.2 Carbon cycle model in JSBACH

We use the DGVM JSBACH (Raddatz et al., 2007; Reick et al., 2013), the land surface model of the Max Planck Insitute Earth System Model (Giorgetta et al., 2013). Vegetation distribution in JSBACH is represented with 12 plant functional types (PFTs), of which 8 are natural types (4 forest types, 2 shrub types, 2 grass types (C3 and C4)), and 4 are anthropogenic

types (C3 and C4 pastures and crops). The PFTs differ with respect to their phenology, albedo and photosynthetic parameters; photosynthesis is based on Farquhar et al. (1980) for C3 plants and Collatz et al. (1992) for C4 plants. The carbon cycle model in JSBACH describes the carbon allocation, the storage in the vegetation and soils, and losses through respiration and natural disturbances. For each PFT, the net primary production (NPP) is allocated to three vegetation carbon pools: the "green pool" containing living tissues, the "reserve pool" containing sugar and starches and the "wood pool" containing woody material. Each of these pools has different turnover rates, influenced by a background natural mortality and foliage losses due to seasonal and climatic influences. The carbon lost from the vegetation pools via turnover goes into the soils in form of litter where it is decomposed. Following LUC, a fraction of the vegetation carbon goes into litter and the other is released directly to the atmosphere. Additionally, carbon can be lost from the vegetation and soil through disturbances in the form of fire and windthrow.

Decomposition of litter in JSBACH is simulated by the YASSO model. YASSO is calibrated globally based on results from litter bag experiments (Tuomi et al., 2008, 2009, 2011) and has been evaluated on site to regional scale (Karhu et al., 2011; Thum et al., 2011; Lu et al., 2013). Decomposition of litter is distinguished in terms of the solubility of litter in four different compounds (acid, water, ethanol and non-soluble hydrolysable pools) and an additional slow decomposing humus pool. Each of these pools has a different decomposition rate derived from the litter bag experiments. The heterotrophic respiration depends on temperature based on a Gaussian model (Tuomi et al., 2008) and on precipitation based on an exponential function (Tuomi et al., 2009). For all PFTs non-woody litter has the same decomposition rates, while the decomposition of woody litter depends on the woody diameter. Additionally, litter is split into aboveground and belowground where the aboveground litter burns while belowground litter does not. All the litter pools–aboveground are belowground–and the humus pool are summed up in obtaining the total soil carbon. YASSO shows a better correlation of present-day carbon stocks with the Harmonized World Soil Data Base compared to JSBACHs' previous soil model CBALANCE (Goll et al., 2015). YASSO has been shown to have a lower sensitivity to some uncertain model parameterizations such as the fraction of carbon lost to the atmosphere following LUC (Goll et al., 2015). A detailed description of the implementation of YASSO can be found in Thum et al. (2011) and in Goll et al. (2015).

## 2.3 Simulation setups

We perform idealized LUCs in which only one vegetation type covers the entire globe and which is subsequently transformed to another type. The idealized simulations approach prevents interference of soil C changes that occur due to different types of LUCs occurring simultaneously in a grid cell or due to sequences of LUC over time. Such interferences occur in realistic LUC simulations. Here, most grid cells in the globe contain a mixture of different vegetation types and at a given year different LUCs may occur. For example, part of the forest in a grid cell may be converted to crop and at the same time part of the grass be converted to crop. Many DGVMs do not separate the soil C for the different PFTs and have one soil C pool for all the PFTs. Those that separate the soil C, e.g. JSBACH, typically add the soil C of the old PFT to the new PFT after LUC. Therefore, soil C change resulting from a specific LUC cannot be obtained using such realistic simulations. The idealized simulations

approach used in this study ensures that starting with equilibrium soil C from one land use then changing to another land use, the resulting soil C change can be associated with the specific LUC.

We create idealized land cover maps for four vegetation types; forest, crop, grass and pasture. In these cover maps the entire globe is covered by each of the four vegetation types. The regions where one of these vegetation types does not exist are masked out in our comparison of simulated results to the meta-analyses (see section 2.4). Each land cover map consists of several PFTs: Forest land cover contains evergreen and broadleaf PFTs in the tropical and extratropical regions, while crop, grass and pasture land cover contains both C3 and C4 PFTs. To create the idealized land cover maps we start with a present day JSBACH land cover map obtained by remapping observed vegetation distribution into PFTs (see Friedl et al. (2010) and supplementary material section S1). In the grid cells where two PFTs belonging to the same vegetation type already exist, e.g., in a grid cell with both tropical deciduous and tropical evergreen from observed vegetation distribution, we scale the cover fraction to the entire grid cells based on their relative distribution.

The carbon cycle model in JSBACH can be executed as part of the entire vegetation model or as a stand-alone model isolating the actual carbon cycle simulation from the simulation of other processes, such as photosynthesis and hydrological processes. In the stand-alone mode, the model is driven by net primary production (NPP), leaf area index (LAI), precipitation and 2 m air temperature together with the vegetation distribution. This setup has the advantage that the model can be run for centennial to millennial timescales at low computational costs.

To obtain the inputs for the stand-alone carbon model, we first perform idealized land-use simulations with JSBACH with each of the four created land cover maps (forest, crop, grass and pasture). We use observed climate from the climate research unit (CRU) for the years 2001 to 2010 as forcing for JSBACH in these simulations (Harris et al., 2014). In a second step, we force the stand-alone carbon cycle model using the NPP and LAI obtained from the JSBACH idealized land-use simulations, precipitation and temperature from CRU, and the idealized vegetation distribution used in the JSBACH simulations. We run the model until the soil C pools are in equilibrium for each of the four land covers. We consider the total soil C in YASSO to be in equilibrium when the relative change in soil C from one year to the next in the grid cell is less than 1%. To perform the LUCs in Table 1, starting from the obtained equilibrium state for each land cover, we use JSBACH land use transition matrices as described in Reick et al. (2013). We modify the transition matrix to perform the respective LUC transition in all the grid cells in the entire globe at the first simulation year with no other LUC transitions during the rest of the simulation time. The distribution of PFTs for the target land cover map is taken from the idealized land cover maps described before, with the exception that the LUC transition to pasture assumes an equal distribution of C3 and C4 pastures (following the default JSBACH assumptions). These simulations represent the standard model version results.

Vegetation productivity as simulated by JSBACH has been shown to be higher as compared to observations (Anav et al., 2013; Todd-Brown et al., 2013). We perform additional simulations where we prescribe observed NPP and LAI instead of using the NPP and LAI simulated by JSBACH. This set of simulations serves two purposes: to assess if the model bias in vegetation productivity has an effect on the soil C response to LUC and to obtain soil C response that is more representative of the observational data in the meta-analyses. In this simulation we use gross primary production (GPP) obtained by extending flux net tower measurements using machine learning algorithms and LAI obtained from MODIS satellite (Tramontana et al.,

2016). The global vegetation classification used for the GPP and LAI data is not the same as the PFTs classification used in DGVMs. We remap the GPP and LAI into JSBACH PFTs; subsequently we derive NPP from GPP (details in supplementary material section S1). We replace the model NPP and LAI with the remapped ones and run the model to equilibrium for the different land cover maps and LUCs.

From the results shown below, we find that one reason for the deviation of simulated soil C response to LUC from the meta-analyses could be the lack of explicitly accounting for crop harvest in the model. To account for the influence of crop harvesting in the model, we introduce a crop harvesting similar to what has been previously done in other DGVMs (Shevliakova et al., 2009; Bondeau et al., 2007; Stocker et al., 2011; Lindeskog et al., 2013). We introduce a harvest pool for the crops that decays into the atmosphere on a timescale of one year. This is in contrast to the earlier model version, where all material harvested from crops was transferred to the litter. In the grid cells with an explicit growing season, harvesting is thereby done at the end of the growing season. In the grid cells without an explicit growing season, as occurs in the humid tropics, harvesting is done constantly throughout the year, imitating that each grid cell contains many individual fields that are harvested at different points in time. 50% of what is harvested is kept in the harvest pool while the other 50% goes to litter. The choice to transfer 50% to the litter is approximated from the average root to shoot ratio of several crop types (Extended data Fig. 2 in Gray et al., 2014). The 50% accounts for root biomass, unharvestable parts of the stem biomass being left in the field and a potential return of carbon to soil in the form of manure.

We perform additional simulations to test the sensitivity of simulated soil C changes towards the representation of natural disturbances, in particular fire. As discussed in section 4, in the standard setup of JSBACH fire affects natural grasslands but not pastures and croplands. Our sensitivity simulations therefore exclude fire on natural grasslands as well. Table 2 summarizes the simulations performed in this study and the names used to represent the respective simulations.

## 2.4 Model-data comparison approach

The idealized LUC simulations represent soil C changes for the entire vegetated areas including regions where LUC does not take place. Therefore, we need a criterion for selecting the model regions to consider in the comparison to the meta-analyses. We select regions in the model based on two different criteria: climate and LUC applied independently. For the climate-criterion, we select the grid cells that fulfill the precipitation and temperature range represented by the meta-analyses in Table 1. Previous studies found that the soil C response to LUC varies spatially due to many factors, among them precipitation and temperature (Don et al., 2011; Wei et al., 2014; Marín-Spiotta and Sharma, 2013). Therefore, the climate-criterion excludes grid cells with different climatic conditions from the meta-analyses, which have potentially different response to LUC. The climate-criterion-based regions are shown in Fig. S1. To assess if the regions obtained using the climate-criterion are representative of regions where the specific LUC has occured historically, we obtain other regions using the differences between present-day and historical land cover in JSBACH. We select the grid cells where more than 10% of the specific vegetation type within the grid cell has undergone LUC. These regions represent the LUC-criterion-based regions (regions in Fig. S2). The results shown in section 3 are averages over the climate-criterion-based regions. We also include a comparison of the simulated changes for these two criteria.

The comparison of soil C changes includes two variables; the relative and absolute soil C changes. We calculate the absolute soil C change by subtracting the soil C of the previous land use from the soil C of the current land use. The relative changes are then calculated with respect to the previous land use. Additionally, we use the generalized CRFs derived from the meta-analyses in Poeplau et al. (2011) to compare the simulated transient response with the meta-analyses. In this case, only the CRFs with high model efficiency for the crop to grass and crop to forest LUCs are used.

The measurements for the individual observations contained in the meta-analyses are done at different ages following LUC. Therefore, the observations may not be in equilibrium for the current land use. To account for this, we sample the simulated soil C changes over the ages represented in the meta-analyses, which makes a direct comparison of the simulated and the observed soil C changes more appropriate. For this we use the age represented by each site in the meta-analyses to select the transient years in the simulations to include in averaging the soil C response. We average the soil carbon response over these years and spatially for the selected regions. This average represents the simulated soil C response over the different ages in the meta-analyses. In section 3 we show both the simulated equilibrium relative and absolute changes and the changes obtained by sampling over the ages represented by the meta-analyses.

## 3 Results

### 3.1 Soil carbon densities for previous and current land use

Before comparing the simulated changes in soil C against the meta-analyses in the next section, we present an assessment of the soil C densities prior to LUC. For this we compare the mean soil C densities in the meta-analyses to the soil C densities for different ecosystems used in bookkeeping models and compiled by the Intergovernmental Panel on Climate Change. For the temperate regions, the previous land use mean soil C of 14.7 kgC m$^{-2}$ for the forests in the meta-analyses (Table 3) is slightly higher than the 13.4 kgC m$^{-2}$ for the undisturbed forest in Houghton et al. (1983), but much higher than the 9.62 kgC m$^{-2}$ in Watson et al. (2000). However, most carbon densities are lower than earlier estimates, such as for tropical forests: 11.7 kgC m$^{-2}$ (Houghton et al., 1983) and 12.27 kgC m$^{-2}$ (Watson et al., 2000); temperate grassland: 18.9 kgC m$^{-2}$ (Houghton et al., 1983) and 23.6 kgC m$^{-2}$ (Watson et al., 2000); and cropland 6-9 kgC m$^{-2}$ (Houghton et al., 1983) and 8 kgC m$^{-2}$ (Watson et al., 2000). A key reason for the lower carbon densities is the limited sampling of only the top-soil in the sites of the meta-analyses (Table 1), while the soil C densities for the different ecosystems in Houghton et al. (1983) and Watson et al. (2000) are up to a depth of 1 m.

The soil C densities in Table 3 obtained at the model simulation depth are much higher compared to the meta-analyses. The lower carbon densities in the meta-analyses are again due to sampling only the top soils. Moreover, the model is in equilibrium for each of the considered land use while the local-scale measurements are done at different times. The average soil C densities for the previous land use in the jsb_drvn simulation are higher than in the t16_drvn simulation for all the LUCs (Table 3). The higher soil C densities result from the generally higher NPP in the jsb_drvn simulation compared to the t16_drvn simulation (Table 4), which in turn leads to higher litter fluxes (Table 5). Accounting for crop harvesting in the jsb_drvn_harv simulations

decreases the litter fluxes (Table 5), which significantly decreases the equilibrium soil C densities. By explicitly accounting for crop harvest in the model the soil C densities for croplands decrease by about 16-24% for the considered regions.

## 3.2 Simulated changes in soil C for the different land-use changes

Figures 1 and 2 show an increase and decrease in soil C following conversion of crop to forest and forest to crop, respectively,
for both the jsb_drvn and the t16_drvn simulations, consistent results from the meta-analyses. In the model this change stems from the higher average productivity in forests compared to croplands for both simulations (Table 4), which leads to higher litter fluxes (Table 5). In addition, woody material in forests decomposes slowly compared to leaf material from croplands. The conversion of crop to grass results in soil C decrease, while the reverse leads to a gain in both of these simulations, which is inconsistent with the meta-analyses (Figs. 1 and 2). The reason for this deviation is related to litter fluxes or process
other than soil decomposition leading to soil C losses, because of observational constraints on the other parts of the carbon cycle model: soil carbon decomposition rates in YASSO are calibrated against a wealth of measurements, and the simulations driven by observation based plant productivity (t16_drvn) result in the same deviation as the JSBACH-driven ones (jsb_drvn). The deviation may stem from an overestimate of cropland relative to grassland litter fluxes, or from an overestimate in the model of non-respiratory processes for grass. Although crop and grass have the same decomposition rates in YASSO, burning
in grasslands leads to the loss of more litter carbon to the atmosphere and shorter turnover time (Table 6). This explains the simulated soil C decrease when croplands are replaced with grasslands. In the jsb_drvn_nofire simulation, switching off disturbances in grasslands leads to model agreement with the meta-analyses on the direction of soil C change (Figs. 1 and 2). The inclusion of crop harvesting in the model reduces the litter fluxes for crops (Table 5) and significantly increases the simulated soil C changes for the different LUCs (Figs. 1 and 2).

Although the simulated equilibrium relative and absolute changes for the conversion of temperate crop to forest and vice versa are larger than in the meta-analyses (Fig. 1), the current land use at the different sites in the meta-analyses may not be in equilibrium. Sampling over the ages represented by the meta-analyses results in relative changes of about 10% for the jsb_drvn simulation and 25% for the t16_drvn simulation for the crop to forest conversion (Fig. 2a). These values are lower compared to the 40% relative changes in the considered meta-analyses and the 53% in Guo and Gifford (2002). For the forest to crop, the
relative changes are about -15% for the jsb_drvn and t16_drvn simulations compared to the -42% in the meta-analyses (Fig 2a). In both of these simulations, the relative changes following the conversion of crop to grass and vice versa are relatively small (Fig. 2a). Despite meta-analyses showing an increase of about 8% for a tropical forest to pasture conversion (Guo and Gifford, 2002; Don et al., 2011), our model results indicate a decrease of about -15%. In addition, the absolute changes are smaller compared to the meta-analyses for all LUCs (Fig. 2b).

Accounting for crop harvesting leads to larger relative and absolute changes in the model. The crop to forest LUC results in an increase of 42%, while the forest to crop results in a decrease of -22%. In line with the meta-analyses, the crop to grass LUC results in an increase of 13%, while the grass to crop results in a decrease of -6% (Fig. 2a). Although these changes are still often smaller than the meta-analyses, they are within the standard deviation represented in the meta-analyses for most of the LUCs (Fig. 2). Comparing the transient response with the generalized CRFs from Poeplau et al. (2011) and the individual

observation points for the crop to grass and crop to forest LUCs, we find that accounting for crop harvesting leads to a stronger soil C response to afforestation in the model and a gain in soil C for the crop to grass conversion, in accordance with the meta-analyses (Fig. 3).

The climate-criterion (temperature and precipitation) used in the selection of the model grid cells for comparison with the meta-analyses leads to small selected regions for the temperate regions (supplementary material Fig. S1). Selecting larger regions based on where the specific LUC has taken place historically, helps in judging if the soil C changes for the climate-criterion are representative of soil C changes in LUC regions (supplementary material Fig. S2). We find that averaging the soil C changes over regions where LUC took place historically results in the same direction of soil C changes as the climate criterion (supplementary material Figs. S3 and S4) with slight differences in the magnitudes of the relative and absolute changes (Fig. 4).

## 4    Discussion

Our results show that the use of meta-analyses provides an opportunity for evaluating simulated soil C response to LUCs. In this section we discuss general issues related to the applicability of meta-analyses for DGVM evaluation, such as scale-related issues, explore the causes of model deviation from the observational data and identify the challenges involved in model-data comparison.

### 4.1    Application of meta-analyses for DGVM evaluation

DGVMs simulate soil C processes at large spatial scales and are widely used to provide soil C estimates relevant for the global carbon budget (Le Quéré et al., 2015). Reliability on these estimates depends on the ability of DGVMs to correctly represent present-day soil C changes from past LUCs. Site-level simulations are often used to evaluate DGVMs for $CO_2$ fluxes, such as net ecosystem exchange and terrestrial ecosystem respiration (e.g., Thum et al., 2011). While vegetation processes representing such variables are well represented in the models, soil processes that are important at the local scale, such as soil chemistry, are not represented in DGVMs. Although it may be impossible for a DGVM to capture the soil C response at an individual site, in particular if the site is not representative of a larger region, the model should be able to match average responses across observations covering a wide region. It is therefore possible to evaluate DGVMs at the scales they are meant for.

In our comparison we choose the grid cells, over which we average the response, based on two independent criteria: the climate space covered by the meta-analyses and the regions where LUC has taken place historically. This selection helps in judging how robust the simulated results are and testing if the meta-analyses are indeed representative of regions where LUC has taken place. Our results for the climate-criterion are qualitatively the same as those of the LUC-criterion (Figs. 1, 2, S3 and S4). Small differences occur for forest to crop and crop to forest in the temperate regions, where the LUC regions have smaller changes compared to the regions captured by the climate-criterion (Fig. 4). This suggests that the regions captured by the meta-analyses by Don et al. (2011) and Poeplau et al. (2011) are generally representative of regions where LUC has taken place historically, although the latter may not be representative of whole-ecosystem averages (see Pongratz et al., 2011). Although

the site studies in the meta-analyses may have biases towards regions of similar soil and climatic conditions (Powers et al., 2011), the mata-analyses still show a large variability compared to our simulated results as indicated by the usually substantially larger standard deviation in the observational data (Fig. 2). This can be explained by the lack of DGVMs in representing the spatial heterogeneity of local soil and climate conditions and land-management practices.

Even though DGVMs provide land-use-related absolute soil C changes, our comparison focused on relative changes. This is the preferred variable in the meta-analyses because spatial heterogeneity partly cancels in relative terms when two sites in close proximity are compared to each other, as done in paired-plots setups. Only relative changes allow for deriving robust carbon response functions (Poeplau et al., 2011). In the jsb_drvn_harv simulation, the equilibrium changes indicate a decrease in soil C of about 11 kgC m$^{-2}$ and 3 kgC m$^{-2}$ for forest to crop and grass to crop, respectively, in the temperate region.

The decrease for forest to crop in the tropics is about 9 kgC m$^{-2}$ (Fig. 1b). The reverse LUCs result in soil C increase of about the same magnitude. Because DGVMs are unaffected by small-scale spatial heterogeneity, their estimates of absolute changes are expected to be more robust than those of meta-analyses and therefore better representative for global C responses. After successful evaluation against relative changes, DGVMs can therefore be used to assess large-scale soil C changes in the absolute terms that are relevant for carbon budget estimates.

## 4.2  Causes of model deviation from meta-analyses

### 4.2.1  Accounting for crop harvesting

The importance of accounting for crop management practices, such as crop harvesting, irrigation and tillage, in DGVMs has been highlighted by recent studies (Levis et al., 2014; Pugh et al., 2015). In particular, Pugh et al. (2015) showed that the inclusion of tillage, grazing and crop harvesting in the LPJ-GUESS model increases the historical land-use carbon emissions.

The increased emissions result from the reduced carbon inputs to the soil by removal of harvested material off-field and increased turnover rates via tillage. Our results show that lack of explicitly accounting for crop harvesting does not only lead to underestimation of soil C changes following the conversion of crop to forest and vice versa, but it also contributes to the wrong direction of change for the crop to grass LUC (Figs. 1 and 2). Figure 3 shows that accounting for crop harvesting in JSBACH improves the temporal response of soil C to the conversion of crop to grass and crop to forest. The removal of

50% crop biomass to the harvest pool–based on root to shoot ratios–is uncertain as it differs across crop types (Table 1.2 in Fageria, 2012); hence this value may not may not be representative of all the sites in the meta-analyses. Despite the uncertainty associated with the harvested crop biomass, our results show that accounting for crop harvesting significantly reduces the soil C for croplands (Table 3).

We note that our model does not represent other crop management practices. For example, tillage in croplands leads to the

exposure of mineral surfaces that are often inaccessible to decomposition causing more soil C loss (Post and Kwon, 2000). However, Pugh et al. (2015) showed that accounting for crop harvesting had larger effects on the historical carbon emissions compared to the inclusion of tillage. Moreover, fertilization can affect cropland soil C stocks by enhancing productivity and hence increasing soil C inputs, and compensating effects by enhancing decomposition by activating microbes (Russell et al.,

2009). The carbon model used in this study simulates soil C based on the plant chemistry and climate. Recent studies have shown that the inclusion of microbial dynamics and priming processes in biogeochemical models can improve model agreement with observations (e.g., Wieder et al., 2013). As these processes are different across land-use types, the inclusion of such processes in future generation of DGVMs may lead to improved simulated soil C response to LUC.

### 4.2.2 Accounting for fire

DGVMs include process representation of vegetation fires to account for the annual emissions of carbon resulting from fires and to allow dynamical shifts in vegetation distribution. However, the choice of which vegetation type burns varies across different DGVMs. Earlier representations of fire in DGVMs accounted for burning only for natural vegetation types (e.g., Kloster et al., 2010; Reick et al., 2013), while recent studies included burning in pastures (e.g., Lasslop et al., 2014) and croplands (e.g., Li et al., 2013). Remote sensing data show that the burned area for different vegetation types varies across different regions. For example, Giglio et al. (2013) showed that while crops contribute to more than 50% of the burned area in Europe and Middle East, grasslands contribute to more than 50% of the burned area in Central Asia. Our model accounts for burning only in natural vegetation types. We perform sensitivity simulations where grasslands are treated the same as croplands by neglecting burning in grasslands in the standard model simulation (jsb_drvn, which does not account for crop harvesting). The sensitivity simulations show a direction of change that is in accordance with the observational data for crop to grass and grass to crop. (Figs. 1 and 2). In the simulations accounting for crop harvesting (jsb_drvn_harv), neglecting burning in grasslands would lead to even larger relative and absolute changes for the crop to grass and grass to crop LUCs. This shows that DGVMs assumptions on which vegetation types burn plays a major role on the soil C response to LUC. However, it remains unclear if the site studies included in the meta-analyses represent regularly burned regions or not. Establishing observational evidence for the sensitivity of soil C changes for a given land use towards frequency and intensity of fire events, similar to how meta-analyses show the sensitivity of responses to factors like precipitation, temperature or soil texture, would allow to evaluate the relevance of this process as currently represented in DGVMs.

### 4.2.3 Conversion of forests to managed grasslands

Results from the meta-analyses have shown that the conversion of forest to pasture in the tropics leads to negligible changes in the soil C and in some cases an increase (Guo and Gifford, 2002; Murty et al., 2002; Don et al., 2011). We find that in the model the conversion of forest to pasture for the tropics leads to a decline in soil C comparable to that of converting forest to crop (Fig. 1). This is associated with larger NPP for forests compared to pastures, which leads to larger litter fluxes (Table 4 and 5). For most of the considered regions in the tropics, the larger simulated NPP for forests compared to pastures is consisted with other observations (Smith et al., 2012). Murty et al. (2002) associated the observed increase in soil C following conversion of forest to pasture with low initial content of soil C, application of fertiliser and avoided grazing. Table 3 shows low previous land use soil C for forest to pasture compared to forest to crop in the meta-analyses. However, the model does not simulate low previous land use soil C for the forest to pasture transition in the considered regions (Table 3).

For the temperate regions, the conversion of grassland to forest increased soil C when the surface litter was included, while without surface litter a decrease in soil C was observed (Poeplau et al., 2011). Our comparison does not include conversions between forest and grass in the temperate regions, the smaller change for grass to crop as compared to forest to crop suggests, however, also here a simulated loss of carbon for the forest to grassland LUC. Schulze et al. (2010) in their review of the European carbon balance found that grasslands store more carbon compared to forests. They attribute this to the higher below-ground allocation for grasslands compared to forests, annual root turnover and possibly nitrogen fixation. Our model does not explicitly represent the potentially deep rooting of grasses, which likely contributes to the disagreement in sign of change for the tropical forest to pasture transition and the weaker simulated response for the temperate grass to crop transition. The latter may further be explained by our simulations not capturing the differences in productivity of grasslands compared to forests and cropland found across various eddy covariance sites in Europe (Schulze et al., 2010). Schulze et al. (2010) found generally larger NPP for grasslands and croplands, while the simulated results shows on average higher productivity for forests for the considered temperate regions (Table 4).

## 4.3 Challenges in model-data comparison

### 4.3.1 Sampling at different times following land-use change

The local-scale measurements constituting the meta-analyses are taken at different times after LUC; hence the current land use is often not in equilibrium. Yet often sites at different stages of disequilibrium are included in average responses, which have been subsequently interpreted in modeling studies as indication for the observation-based evidence of effects of historical land-use change on equilibrium soil C stock changes (Pongratz et al., 2009; Reick et al., 2010; Stocker et al., 2011). Idealized simulations such as presented here can account for this transience in soil C response by sampling over the same ages as represented by the meta-analyses. Due to the larger availability of sites that have recently undergone LUC, averaging over all available sites of different ages in the meta-analyses has a strong bias towards smaller soil C changes than would be expected in equilibrium. In our model, this bias becomes apparent in the smaller relative and absolute changes compared to the equilibrium changes (Figs. 1 and 2). The bias can be quantified in our simulations and amounts to about 20-40% of the equilibrium response that is captured by an average across the simulations accounting for crop harvest (supplementary material Table S4). Therefore, parametrization and evaluation of DGVMs using meta-analyses needs to account for the transient state of the mean soil C changes for the different LUCs represented in the meta-analyses.

### 4.3.2 Different soil sampling depths

Soil carbon models used in DGVMs typically simulate soil processes up to a depth of 1 m and are meant to capture the complete soil C stock changes after LUC. By contrast, some of the observations, in particular in the tropics, covered only a shallow sampling depth (Table 1). Analysis of the depth-dependence of observed soil C changes revealed that most of the change occurs in the top 30 cm (Poeplau and Don, 2013), in line with the fact that in most ecosystems the majority of soil C is stored in the upper layers with around 1520 Pg C, which is more than 56% of the total soil C globally, in the upper 1 m

(Jobbágy and Jackson, 2000). For comparison of the relative and absolute of soil C changes at consistent depth, scaling of the site studies to the model depth can be applied (Yang et al., 2011; Deng et al., 2014). However, these scaling approaches used an equation calibrated across a wide range of ecosystems and are thus independent of the land use types. Hence, such as a scaling would only affect the comparison of the absolute changes, but not the relative changes.

Previous studies have shown that the amount of soil C varies with depth differently in different ecosystems. For example, Jobbágy and Jackson (2000) found that 42% of the total soil C in grasslands is stored in the upper 20 cm while for forests 50% of the carbon is in the upper 20 cm. Guo and Gifford (2002) argued that while forests have high above-ground inputs in the top layers, tree roots are less important sources of organic matter because much of the tree root systems lives for many years. On the other hand, the annual root turnover in grasslands contributes to larger soil C storage in deeper depths. Therefore also the changes of soil C vary with depth differently for different LUCs. Poeplau and Don (2013), using several local scale measurements, found that 91% of the total soil C change occurs in the upper 30 cm following afforestation, while 65% of the change occurs in the top soil following the conversion of crop to grass. In line with this, DGVMs may need to consider including vertically resolved soil profiles to represent the distribution of soil C with depth across different ecosystems, to represent that different types of LUCs act differently depending on the sampled depth and to be better comparable with meta-analyses. Conversely, to capture the full impacts of LUC on soil C as relevant for carbon budgeting and to allow a direct comparison to DGVMs, local-scale measurements need to consider a deeper sampling of the soil profile.

## 5 Conclusions

Our comparison used a typical DGVM, JSBACH, which has been applied in a range of model intercomparison projects (e.g. Brovkin et al., 2013; Le Quéré et al., 2015). It revealed successful representation of some, but not of all LUCs. The comparison supports previous studies that found that inclusion of crop harvesting is a crucial component in DGVMs to accurately represent soil carbon losses with agricultural expansion and historical land-use emissions (Stocker et al., 2011; Pugh et al., 2015). Additionally, we find that natural disturbances by fire, which are not well documented in the meta-analyses, may substantially influence the soil carbon response to LUC simulated by models.

Challenges for this comparison remain. First, meta-analyses cover many observations where the current land use may not be in equilibrium; hence the mean relative changes in the meta-analyses represent a transient response. Idealized LUC simulations can account for this by sampling over the ages represented by the meta-analyses. Second, the meta-analyses include local-scale observations that are done at different sampling depths. Ultimately this challenge can be overcome only by deeper sampling in observational data or by DGVMs considering in the future including a vertically resolved soil profile.

Despite such challenges, our study shows that the use of the meta-analyses on soil carbon changes following LUC offers the opportunity for evaluation and improvement of DGVMs. We developed a systematic approach that is applicable to any DGVM for comparing simulated soil carbon changes due to different LUCs using the meta-analyses. Extending this comparison to other DGVMs or to model intercomparison projects would not only provide an observational reference for validation, but also help investigate across a larger range of processes the key influences on models' sensitivity to LUC.

*Acknowledgements.* We thank Jari Liski for his helpful discussions during the course of this study and Daniel Goll for his contribution in the implementation of the YASSO model into JSBACH. We thank Martin Jung and Ulrich Weber for providing us with the GPP and LAI data used in some of the simulations. We thank Goran Georgievski for his comments, which helped to improve the manuscript. This work was supported by the German Research Foundation's Emmy Noether Program (PO 1751/1-1).

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

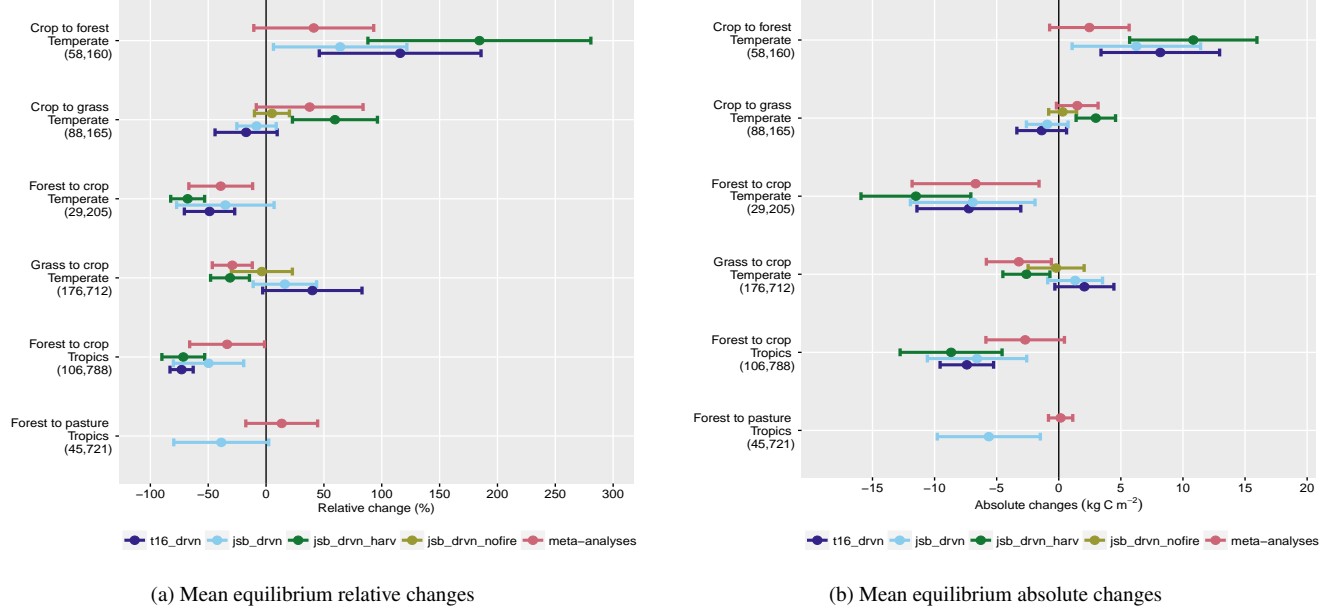

(a) Mean equilibrium relative changes

(b) Mean equilibrium absolute changes

**Figure 1.** Mean simulated equilibrium relative (a) and absolute changes in soil carbon (b) compared to results from the meta-analyses. The first number in the parenthesis represents the number of studies in the meta-analyses and the second is the number of grid cells from the global simulation that fulfill the climate-criterion in the meta-analyses (regions in supplementary material Fig. S1). The dots represent the mean changes and the bars represent the standard deviation.

**Table 1.** Mean annual temperature (MAT) range, mean annual precipitation (MAP) range, mean sampling depths ($\pm$ std) and the mean current land-use age for the local-scale observations in the meta-analyses. We note that the different equilibrium results presented below, e.g., for crop in the crop to forest LUC and the forest to crop LUC, are due to the different climate-criterion (precipitation and temperature) for the different LUCs.

| Land-use change | MAT (°C) | MAP (mm) | Sampling depth (cm) | Age (years) |
|---|---|---|---|---|
| Crop to forest (temperate) | 5.9–10.7 | 540–1020 | 39.53±24.8 | 40.28 |
| Crop to grass (temperate) | 6.7–11.2 | 440–1030 | 23.44±10.5 | 21.7 |
| Forest to crop (temperate) | 3.4–16.4 | 690–1320 | 28.48±13.5 | 50.21 |
| Grass to crop (temperate) | 1–12.7 | 150–960 | 27.11±11.1 | 39.69 |
| Forest to crop (tropics) | 15–27.5 | 570–3400 | 17.5±12.81 | 22.5 |
| Forest to pasture (tropics) | 18–28 | 570–4000 | 15.79±11.55 | 20.67 |

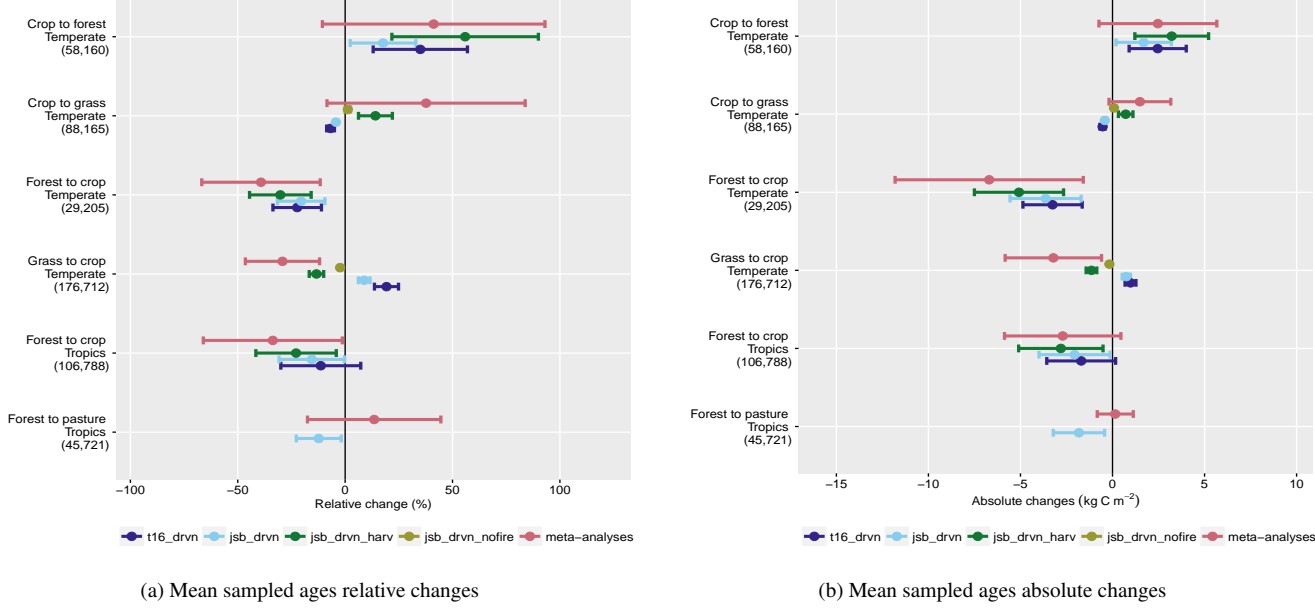

(a) Mean sampled ages relative changes

(b) Mean sampled ages absolute changes

**Figure 2.** Mean simulated relative (a) and absolute changes in soil carbon (b) over the sampled ages represented by the meta-analyses compared to results from the meta-analyses. The first number in the parenthesis represents the number of studies in the meta-analyses and the second is the number of grid cells fulfilling the climate-criterion in the meta-analyses (regions in supplementary material Fig. S1). The dots represent the mean changes and the bars represent the standard deviation.

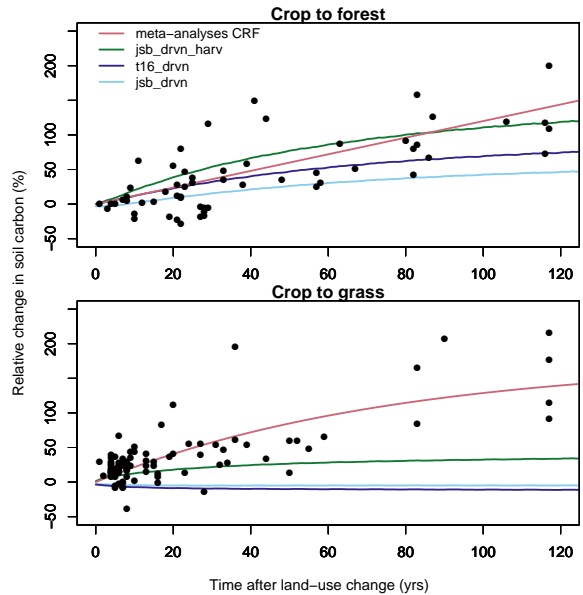

**Figure 3.** Mean simulated transient relative changes in soil carbon compared to the individual observations in the meta-analyses (black dots) and generalized carbon response functions (CRF) as in Poeplau et al. (2011) for the crop to grass and crop to forest LUCs.

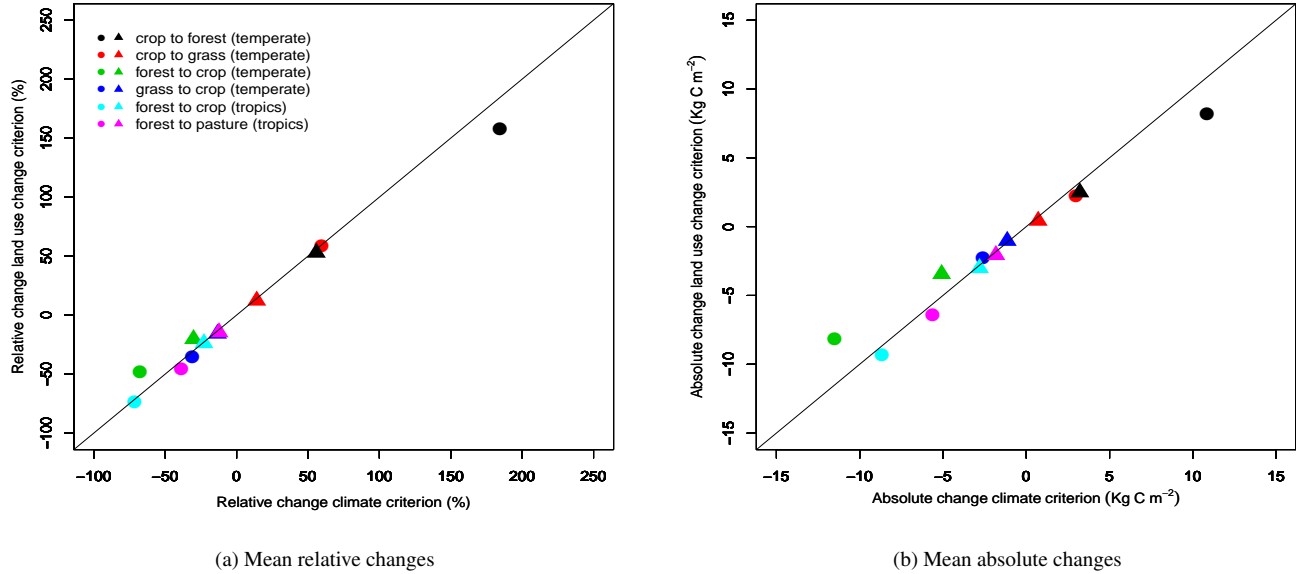

(a) Mean relative changes

(b) Mean absolute changes

**Figure 4.** Mean relative (a) and absolute (b) changes for the different land-use transitions with regions based on the climate (temperature and precipitation) criterion and based on where land-use change has taken place historically for the jsb_drvn_harv simulation. The triangles represent the mean changes over the sampled ages while the circles represent the mean equilibrium changes.

**Table 2.** Summary of the simulations done in this study.

| Simulation name | NPP& LAI | Land-use change | Disturbances | Crop harvest |
|---|---|---|---|---|
| jsb_drvn | Simulated by JSBACH | crop to forest, forest to crop, crop to grass, grass to crop, forest to pasture | on | none |
| t16_drvn | Prescribed from observations | crop to forest, forest to crop, crop to grass, grass to crop | on | none |
| jsb_drvn_harv | Simulated by JSBACH | crop to forest, forest to crop, crop to grass, grass to crop | on | included |
| jsb_drvn_nofire | Simulated by JSBACH | crop to grass, grass to crop | off | none |

**Table 3.** Mean simulated equilibrium soil C densities at the model depth (100 cm) and the mean soil C in the meta-analyses in kgC m$^{-2}$ for previous and current land use in the different LUCs and simulations ($\pm$std). The meta-analyses soil C densities represent the mean over sites with different measuring depths.

| Land-use change | meta-analyses | | t16_drvn | | jsb_drvn | | jsb_drvn_harv | |
|---|---|---|---|---|---|---|---|---|
| | Previous | Current | Previous | Current | Previous | Current | Previous | Current |
| Crop to forest (temperate) | 6.8±3.1 | 9.3±5.1 | 7.2±1.7 | 15.4±5.5 | 10.1±2.9 | 16.9±6.4 | 6.0±2.9 | 16.9±6.4 |
| Crop to grass (temperate) | 4.6±2.1 | 6.1±2.6 | 7.4±1.7 | 6.0±2.1 | 9.5±2.6 | 8.6±2.8 | 5.6±2.6 | 8.6±2.8 |
| Forest to crop (temperate) | 14.7±5.3 | 8.0±2.7 | 13.4±5.1 | 6.1±1.7 | 16.5±5.4 | 8.4±2.7 | 16.5±5.4 | 5.0±1.8 |
| Grass to crop (temperate) | 11.5±6.7 | 8.3±5.6 | 6.2±2.1 | 8.3±2.9 | 8.4±3.9 | 9.8±4.7 | 8.4±3.9 | 5.7±2.8 |
| Forest to crop (tropics) | 6.4±3.9 | 3.7±2.3 | 10.1±2.4 | 2.7±1.1 | 11.4±4.3 | 4.8±1.7 | 11.4±4.3 | 2.7±1.1 |
| Forest to pasture (tropics) | 3.7±2.8 | 3.9±2.6 | - | - | 11.4±4.1 | 5.8±1.8 | 11.4±4.1 | 5.8±1.8 |

**Table 4.** Mean annual NPP for previous and current land use in kgC m$^{-2}$ for the different LUCs and simulations ($\pm$ std).

| Land-use change | t16_drvn | | jsb_drvn | |
|---|---|---|---|---|
| | Previous | Current | Previous | Current |
| Crop to forest (temperate) | 0.42±0.10 | 0.73±0.24 | 0.58±0.15 | 0.90±0.34 |
| Crop to grass (temperate) | 0.43±0.09 | 0.41±0.14 | 0.57±0.15 | 0.63±0.17 |
| Forest to crop (temperate) | 0.77±0.26 | 0.44±0.12 | 1.04±0.34 | 0.58±0.14 |
| Grass to crop (temperate) | 0.32±0.11 | 0.34±0.12 | 0.48±0.24 | 0.44±0.23 |
| Forest to crop (tropics) | 1.21±0.28 | 0.35±0.10 | 1.42±0.60 | 0.69±0.22 |
| Forest to pasture (tropics) | - | - | 1.46±0.59 | 0.87±0.20 |

**Table 5.** Mean annual equilibrium litter fluxes in kgC m$^{-2}$ for previous and current land use in the different LUCs and simulations ($\pm$ std).

| Land-use change | t16_drvn | | jsb_drvn | | jsb_drvn_harv | |
|---|---|---|---|---|---|---|
| | Previous | Current | Previous | Current | Previous | Current |
| Crop to forest (temperate) | 0.41±0.10 | 0.66±0.21 | 0.57±0.14 | 0.79±0.28 | 0.35±0.14 | 0.79±0.28 |
| Crop to grass (temperate) | 0.41±0.09 | 0.39±0.13 | 0.55±0.14 | 0.58±0.15 | 0.34±0.14 | 0.58±0.15 |
| Forest to crop (temperate) | 0.74±0.25 | 0.44±0.11 | 0.95±0.32 | 0.58±0.13 | 0.95±0.32 | 0.34±0.09 |
| Grass to crop (temperate) | 0.30±0.10 | 0.33±0.10 | 0.44±0.21 | 0.43±0.23 | 0.44±0.21 | 0.26±0.14 |
| Forest to crop (tropics) | 1.21±0.28 | 0.35±0.10 | 1.28±0.54 | 0.63±0.19 | 1.28±0.54 | 0.37±0.12 |
| Forest to pasture (tropics) | - | - | 1.31±0.53 | 0.78±0.16 | 1.31±0.53 | 0.78±0.16 |

**Table 6.** Mean soil carbon turnover time (years) for the previous and current land use for the jsb_drvn simulation with and without disturbances.

| Land-use change | Previous | Current |
|---|---|---|
| Crop to grass, with disturbances | 17.1±4.5 | 15±2.6 |
| Crop to grass, no disturbances | 17.1±4.5 | 17.2±4.3 |
| Grass to crop, with disturbances | 21.9±8.3 | 28.7±14.9 |
| Grass to crop, no disturbances | 28.6±14.5 | 28.5±14.5 |