# Peer review of "Soil carbon response to land-use change: Evaluation of a global vegetation model using observational meta-analyses"

_Biogeosciences, 2016_

## Referee Comment (RC1) · B. Guenet (Referee) · 28 Jun 2016

The study by Nyawira et al., it is nice attempt to evaluate a large-scale model using meta-data. This study focus of LUC effect on SOC dynamic but the methodology presented here might probably use in another context (compare long term and short term effect of atmospheric CO2 increase on NPP for instance).

The paper is generally well written but the methods section needs to be a bit more detailed to be useful to any modeller interested in applying the method. In particular, how the idealized simulations were sampled for non-equilibrium cases. Another missing point is how tillage is represented in the model and in particular its effect on SOC.

[Figure]

The take home message I found in the paper is that using observed GPP and with harvest representation the model fits better with the data. These results are not very surprising except if different approaches has been tested but not presented. Nevertheless, the main interest of the paper to my opinion is methodological. Therefore I suggest to add the scripts used in supplementary material to facilitate the use of the method by other.

Finally I suggest accepting the paper with minor revisions.

Minor comments: P4 l4: I don't understand this part. If you did idealized simulation using one vegetation type per grid, why give those details about grid cells with more than one vegetation type?

P4 l16: If this is the case here the word "usually" is not necessary.

P4 l29: When LUC is performed it is not very clear how the new vegetation type is split into the different PFTs?

P5 l6: The product of degradation of this new pool goes back to litter (to simulate composting for instance) or is this OM totally exported?

P6 l4: It is quite a big assumption to fix this ß value since it is likely controlled by several factors (Matthieu et al., 2015). A sensitivity analysis to this parameter might be useful in the supplementary materials.

Fig. 2: In the title: it is not "equilibrium" anymore right?

Tab. 4: It seems that to force the model with observed GPP and to better reproduce harvest improved the model-data agreement, what about doing both?

Tab. 5: Comparison with data might be useful in particular to see the error associated to autotrophic respiration in the model.

---

## Referee Comment (RC2) · D. Schepaschenko (Referee) · 3 Jul 2016

The study demonstrates an approach for evaluating performance of DGVMs to account soil carbon changes following land use change. It is important to estimate how far DGVM simulations are from the reality and which model setup is closer to observations. The article has rich discussion section, where most of the issues are covered. The paper is well written. Finally I suggest accepting the paper with minor revisions.

I have two concerns:

1. Application of universal function for scaling soil carbon pool to 100 cm (page 3, line 15) could introduce substantial bias. In many cases carbon pool changes in top

layer only, but you propagate observed value down to 1m and increase therefor the magnitude. That might be the reason of having higher amplitude in meta-data. I would either suggest use soil map and soil specific equations or make the analysis for the top layer only if no observation for deep soil layers available.

2. Selection of area/climate for simulation is important. By including extra area (where LUC not going to happen) or excluding potential LCC area, you might bias the overall estimation. Authors suggest three different extends and each is not ideal:

- Entire vegetated area of the land surface (too big. Low chances e.g. for forest on permafrost to be converted to cropland)

- Area where LUC has taken place historically (too narrow, LUC might came to new places, e.g. tropical deforestation)

- Where meta-data were available (even more narrow) I would suggest to overlay PFT map with climate one and define climatic patterns where one or another PFT can appear. This would cover all current and potential LUC.

Minor comments:

Page 7, line 8. In fact, forest might have lower NPP compare to cropland, but most of the dead matter come to soil surface where decomposition is slow.

Page7, line 10. Here could be different explanation. Soil respiration is higher in cropland compare to grassland because of the tillage. That is why having similar NPP grassland accumulate more carbon.

Page 9, line 10 "Without accounting for crop harvesting" makes sense only to demonstrate how DGVMs are far from reality while not taking into account such evident things like management and disturbances.

Page 9, line 10 "switching off disturbances in grass ... leads to the right direction of soil carbon change" hope we aim is to describe the reality with the model, but not just

have a similar estimation. Disturbances exist. If result is better without disturbances, then the model makes mistake in its different part.

Page 10, Line 1. Grassland and cropland NPP generally larger compare to forest in temperate region also because they allocated on best locations (soil, slope, etc.). However if you try to convert existing forest to grassland or cropland you might not get increase of NPP.

---

## Referee Comment (RC3) · E. Marín-Spiotta (Referee) · 7 Jul 2016

This study uses published results from temperate and tropical meta-analyses that calculate mean responses of soil carbon change to different land-use transitions from field measurements of paired plots to better constrain estimates of belowground response to land-use change at a global scale simulated by a dynamic global vegetation model. To my knowledge, this is the first time global syntheses of data from published meta-analyses have been used to compare to results from models. The research takes advantage of a large effort to synthesize global temperate and tropical data on soil C to estimate the response of soil C stocks to major land use transitions.

Overall, the writing could be improved to more clearly describe the modeling approach

and to distinguish it from past efforts. Some sections have minimal text (for example, description of the observational data and the meta-analysis approach), and while the attempt to be concise is appreciated, more information would make it easier for readers to understand and attempt to replicate the approach.

The discussion dives directly into details of the model but would benefit from an overall summary highlighting the main findings of the paper and being organized around the take home messages of the research. An effort to frame the discussion in a bigger context will help identify novel insights to a broader audience who may not be familiar with the modeling approach but is still very interested in the findings. For example, consider starting the discussion with section 4.1.4 (which has a great discussion of scale between the models and the field observations) then going into the details of the crop harvest and fire, then discussion of the challenges (current section 4.2.1).

The use of the term "meta-data" in the title and throughout the paper to represent results from a meta-analysis (a specific statistical test that calculates differences (effect size or response ratio) between data points) is confusing and inaccurate (and distracting) as this term has a different formal definition ("data that describes other data"). To avoid unnecessary confusion, please use an alternate term, such as "field data," "observations", "observational data" or "results from meta-analyses." The term meta-data in the paper is used to refer to meta-analyses (published studies using the specific statistical approach), to the results of these analyses and to general syntheses of published data, further adding to confusion, as these are not the same.

In a few places, it is unclear how data used in the model simulations is then related to the land cover types and information associated with the observations of soil carbon change. For example, how are the different plant functional types, especially the different forest PFT, related to the 4 idealized land use classes? Given that the observational data used in this paper is heavily biased towards tropical sites (from Don et al. 2011 vs Poeplau and Don 2015), it is expected that the land cover description of the sites in the published literature do not match the PFTs at the global scale in the

DGVM. In addition, where do the plant productivity measurements used in the model come from and how do these relate to the types of vegetation and their growth rates from the observational soil carbon studies?

How do the model simulations address uncertainties in field soil C measurements? For example, the values given in section 3.1 are averages with some associated error. What is the size of this error, how does this variability affect the carbon response functions, and how then do these influence modeled results?

P. 1, l. 23: Soil C changes with LUC are not only influenced by differences in inputs, but also outputs, and alteration to processes that store C in soils. The last paragraph of section 4.1.1 in the discussion briefly starts to address other factors that can influence soil C decomposition that are not included in the model. Some further discussion on how the focus on plant litter chemistry and climatic variables as controls on C cycling that is the basis for the biogeochemical component of the model and the absence of other mechanistic controls on soil C turnover and how they may influence differences between simulation results and observational data would enhance the paper.

P. 4, l.13-20: This paragraph discusses grid cells with only one vegetation type and also proportions of grid cells undergoing different land use transitions from different vegetation types. Please clarify which approach was taken in the paper and distinguish between old approaches (for example, additive soil C pools with LUC) and the new one proposed in this study.

P. 5, l. 3: What are the four idealized land use cases? These could be identified here or earlier in the description of the observational data and the meta-analyses.

Minor comments

P. 1, l. 5-11: Consider the following sentence reorganization: "Our simulated results show model agreement with observational data on the direction of changes in soil carbon for some land-use changes, although the models generally estimated smaller

magnitudes of change. The conversion of crop to forest resulted in a simulated soil carbon gain of 10% compared to a gain of 42% in the data, whereas the forest to crop change resulted in a simulated loss of-15% compared to -40%. The model and field data disagreed for the conversion of crop to grassland. The simulations estimated a small soil carbon loss (-4%), while field data indicate a 38% gain in soil C with the same land-use transition. These model deviations from the observations are substantially reduced by explicitly accounting for crop harvesting and removing burning in grasslands from the model."

P. 1, l. 17: suggest deleting: "(hereafter meta-data)" as this is an incorrect use of this term

P. 1, l. 18: add references to the meta-analyses here

P. 2, l. 7: Rewrite: "Despite the dependence of the soil carbon response to local conditions of soils, climate and management practices, regional and global syntheses of published data can be useful to aggregate local-scale measurements on soil carbon changes and estimate mean responses to different LUCs using a meta-analysis approach."

P. 2, l. 8 and 10: Here meta-data should be replaced by meta-analyses.

P. 2, l. 10 and P. 3, l. 7: what is meant by "harmonize"a temperate ? Can you use another term?

P. 2, l. 15: Marin-Spiotta and Sharma (2013)'s work did not use a meta-analysis approach

P. 3, l. 5: replace "the meta-data" with "results from the meta-analyses"

P. 3, l. 6: The "quality criteria" sentence structure is awkward. Consider: "These meta-analyses were conducted on paired plots of similar soil type and texture, to reduce uncertainties from hetereogeneous soil properties unrelated to the land-use transition."

P. 3, l. 30: Replace windbreak (check definition) with windstorm.

P. 4, l. 3-5: Consider rewording: "The decomposition rate of litter is controlled by its chemical composition, as determined by its solubility (acid, water, ethanol and non-soluble hydrolysable pools) and the presence of a slow decomposing humus pool." It is unclear from the text whether the humus pool is part of the plant litter or a soil organic matter pool? Does the model include above and belowground litter pools?

P. 4, l. 26: replace "ran" with "run"

P. 10, l. 6: above refers to what?

Figures and Tables

Figures 1 and 2 are hard to read. Consider that the green, orange and brown colors will be difficult to distinguish for readers with color-blindness, which affects almost 10% of males in many European and English-speaking countries. The grey background also reduces the contrast between the lines, and the lines are too small and hard to see.

Figure 3. See earlier comment about choice of colors.

---

## Referee Comment (RC4) · T. Pugh (Referee) · 7 Jul 2016

Nyawira et al. develop a framework to evaluate the response of soil carbon stocks in the DGVM JSBACH to land-use change, using meta-analyses of observations of soil carbon stock change. They find that the baseline model is unable to reproduce the observations for most transition types tested, but that inclusion of crop harvest and exclusion of fires on pasture land notably improves the fit of the model response.

The analysis has been carefully executed and the manuscript is well written. It provides a useful framework for evaluation of soil carbon response to land-use change, which is generally poorly evaluated in DGVMs used to provide land-use emission estimates, despite constituting a substantial part of the overall emission. I am happy to

recommend publication, subject to addressing the following minor comments.

Section 2.2 - In order to understand the differences between the various land-use types considered, more information about the different PFT types is required. In particular, how do C3/C4 grasses, C3/C4 pastures and C3/C4 crops differ from one another? I suggest to add a table listing the differences in any important PFT parameters, or other parameters which may be important to the land-use type (e.g. different soil decomposition rates?).

Pg. 4, line 21 - I wasn't quite sure here if the forest PFTs had been extended to cover areas not presently covered by forest, or not. Can the authors clarify?

Pg. 5, line 12 - 50% seems a high proportion of crop NPP to be allocated below-ground. This will have a substantial influence on the size of the flux lost to harvest and should therefore be discussed in relation to published literature in the discussion section.

Pg. 5, line 27 - I think these are grid-cells where just the relevant transition type has taken place (based on Fig. S2), and not where any LUC has taken place at all? This isn't clear to me from the text. Also, on first reading I thought climate and LUC criterion were being applied simultaneously, and it only later became clear that they were being applied separately.

Pg. 6, line 4 - Should beta have units of length? Also, please define the units of $d_0$ (presumably cm).

Pg. 6, line 13 - How do you sample simulated soil C changes over the ages? Do you take a simple mean over the age range in the observations, or do you weight the mean by the number of observations in each age range? I would argue the second is much better, if you have the data to do it.

Section 3.2, para. 1 - I think you can be a bit more assertive here in saying that the reason for the results from the crop to grass simulations is fire. That seems to be very

clearly demonstrated at the end of the paragraph, and I'm not sure why the section beginning "we suspect" (line 11) is included.

Pg. 8, line 14 - Should Fig. S2 be cited here?

Pg. 9, line 24 - What is meant specifically by "forest floor"? Surface litter?

Pg. 9, line 27 - Is the larger NPP for forests than pastures in accordance with the literature? Would be good to discuss this briefly with some references.

End of section 4.1.4 - Absolutely agree with this sentiment, but shouldn't we then be aiming for a more stringent test than getting within the very large standard deviation that results from this small-scale heterogeneity?

Pg. 11, line 23 - What exactly is meant by "top soil"?

Section 4.3 - I agree with the general statement regarding absolute estimates, but the way this section is written seems to imply that JSBACH was successful in capturing the observations in this evaluation, which I feel would be stretching it a bit for several of the transition types, especially grass to crop (based on Fig. 2).

Table 4 - I'm not clear on the logic of having this table in addition to Table 3. It would seem more helpful to add the obs_drvn and jsbach_drvn_harv data to Table 3 (appropriately adjusted for 30 cm depth), to allow them to also easily be assessed against the observations.

---

## Author Comment (AC1) · 24 Aug 2016

[twoside]article

lipsum [sc]mathpazo amssymb, amsmath graphicx [T1]fontenc microtype [final]pdfpages [hmarginratio=1:1,top=32mm,columnsep=20pt,left=26mm]geometry multicol [hang, small,labelfont=bf,up,textfont=it,up]caption subcaption [subfigure]subrefformat=simple,labelformat=simple,listofformat=subsimplebooktabs float hyperref lettrine paralist abstract titlesec

fancyhdr

**Response to B. Guenet of the paper entitled "Soil carbon response to land-use**

**change: Evaluation of a global vegetation model using observational meta-analyses"**

Ref.: bg-2016-161

Below are the reviewers suggestions (***bold italic font***) and our responses to each point (normal font). In some of our responses, we have cited text from the revised manuscript (*italic font*).

***The study by Nyawira et al., it is nice attempt to evaluate a large-scale model using meta-data. This study focus of LUC effect on SOC dynamic but the methodology presented here might probably use in another context (compare long term and short term effect of atmospheric CO2 increase on NPP for instance).***

Thank you for your comments. We are happy that you find our method useful for other applications.

***The paper is generally well written but the methods section needs to be a bit more de- tailed to be useful to any modeller interested in applying the method. In particular, how the idealized simulations were sampled for non-equilibrium cases.***

We have expanded the simulation setups section (section 2.3) and the model-data comparison approach section (section 2.4) to make the method easily understandable to readers interested in doing similar analysis.

***Another missing point is how tillage is represented in the model and in particular***

*its effect on SOC.*

The model does not represent other crop management practices such as tillage. We discuss this and the implications of tillage for SOC in section 4.2.1.

*The take home message I found in the paper is that using observed GPP and with harvest representation the model fits better with the data. These results are not very surprising except if different approaches has been tested but not presented. Nevertheless, the main interest of the paper to my opinion is methodological. Therefore I suggest to add the scripts used in supplementary material to facilitate the use of the method by other.*
*Finally I suggest accepting the paper with minor revisions.*

The scripts and the data used for our analysis are archived by the institute and are available upon request, in accordance with the guidelines of good scientific practice of the Max Planck Society. We have added this information in the acknowledgement.

*Minor comments: P4 l4: I don't understand this part. If you did idealized simulation using one vegetation type per grid, why give those details about grid cells with more than one vegetation type?*

In this paragraph we describe why we use idealized simulations and also why realistic LUC simulations with a mix of vegetation types cannot be used for evaluating DGVMs. We have re-written the paragraph to make this point clear.
*"We perform idealized LUCs in which only one vegetation type covers the entire globe and which is subsequently transformed to another type. The idealized simulations approach prevents interference of soil C changes that occur due to different types of LUCs occurring simultaneously in a grid cell or due to sequences of LUC over time. Such interferences occur in realistic LUC simulations. Here, most grid cells in the globe contain a mixture of different vegetation types and at a given year different LUCs*

*may occur. For example, part of the forest in a grid cell may be converted to crop and at the same time part of the grass be converted to crop. Many DGVMs do not separate the soil C for the different PFTs and have one soil C pool for all the PFTs. Those that separate the soil C, e.g. JSBACH, typically add the soil C of the old PFT to the new PFT after LUC. Therefore, soil C change resulting from a specific LUC cannot be obtained using such realistic simulations. The idealized simulations approach used in this study ensures that starting with equilibrium soil C from one land use then changing to another land use, the resulting soil C change can be associated with the specific LUC."*

**P4 I16: If this is the case here the word "usually" is not necessary.**

We have removed the sentence.

**P4 I29: When LUC is performed it is not very clear how the new vegetation type is split into the different PFTs?**

We have expanded the description of the initial PFT distribution (paragraph 2 of section 2.3) as well as the changes in PFT distribution with LUC (paragraph 4 of section 2.3).
*" To perform the LUCs in Table 1, starting from the obtained equilibrium state for each land cover, we use JSBACH land use transition matrices as described in Reick et al. (2013). We modify the transition matrix to perform the respective LUC transition in all the grid cells in the entire globe at the first simulation year with no other LUC transitions during the rest of the simulation time. The distribution of PFTs for the target land cover map is taken from the idealized land cover maps described before, with the exception that the LUC transition to pasture assumes an equal distribution of C3 and C4 pastures (following the default JSBACH assumptions). These simulations represent the standard model version results."*

**P5 I6: The product of degradation of this new pool goes back to litter (to simulate**

*composting for instance) or is this OM totally exported?*

The harvest pool decays solely into the atmosphere within one year. Additional organic matter that may be transported back to the field in form of manure is captured implicitly by the biomass left in the field after harvest. We have added this in the text.

*P6 l4: It is quite a big assumption to fix this $\beta$ value since it is likely controlled by several factors (Matthieu et al., 2015). A sensitivity analysis to this parameter might be useful in the supplementary materials.*

We have removed the section on scaling soil carbon in the manuscript.

*Fig. 2: In the title: it is not "equilibrium" anymore right?*

We have changed the figure caption.

*Tab. 4: It seems that to force the model with observed GPP and to better reproduce harvest improved the model-data agreement, what about doing both?*

This is a good suggestion. However, we did not perform this simulation due to technical issues related to how we trigger the harvest events based on phenology in the model. Harvest in the regions with a well defined growing season (e.g., temperate regions) is done at the end of the growing season. In the observation driven simulation, the prescribed LAI seasonality had drops during the growing season that would lead to constant harvesting during the growing season, which would introduce an artificial bias between the model-driven and the observation-driven simulations. However, since the observation-driven and jsbach-driven simulations results for the different LUCs were similar this simulation would not change the conclusions in the paper.

*Tab. 5: Comparison with data might be useful in particular to see the error asso-*

***ciated to autotrophic respiration in the model.***

The NPP values shown in Table 5 are model inputs and not outputs. The obs_drvn simulation values represent the NPP obtained from GPP using ratios. We have added a discussion in the supplementary to discuss uncertainties associated with scaling the GPP to NPP.

---

## Author Comment (AC2) · 24 Aug 2016

[twoside]article

lipsum [sc]mathpazo amssymb, amsmath graphicx [T1]fontenc microtype [final]pdfpages [hmarginratio=1:1,top=32mm,columnsep=20pt,left=26mm]geometry multicol [hang, small,labelfont=bf,up,textfont=it,up]caption subcaption [subfigure]subrefformat=simple,labelformat=simple,listofformat=subsimplebooktabs float hyperref lettrine paralist abstract titlesec

fancyhdr

**Response to D. Schepaschenko of the paper entitled "Soil carbon response to**

**land-use change: Evaluation of a global vegetation model using observational meta-analyses"**

Ref.: bg-2016-161

It seems our point by point responses at the quick report stage of the manuscript unfortunately were not passed to the reviewer. The requested changes were accounted for before the paper was published on discussions.

Below are the reviewers suggestions (***bold italic font***) and our responses to each point (normal font) based on our changes prior to publication as discussion paper and additional changes in response to the other reviewers' comments.

***This study demonstrates an approach for evaluating perfomance of DGVMs to account soil carbon changes following land-use change. It is important to estimate how far DGVM simulations are from the reality and which model setup is closer to observation. The article has rich discussion section, where most of the questions are covered. The paper is well written. Finally I suggest accepting the paper with minor revisions.***

***Application of universal function for scaling soil carbon pool to 100 cm (page 3, line 15) could introduce substantial bias. In many cases carbon pool changes in top layer only, but you propagate observed value down to 1m and increase therefor the magnitude. That might be the reason of having higher amplitude in meta-data. I would either suggest use soil map and soil specific equations or make the analysis for the top layer only if no observation for deep soil layers***

*available.*

The reviewer is correct that the scaling of the meta-data with depth is quite uncertain. We didn't find reliable land-use specific functions for scaling soil carbon densities. Therefore, we have removed the scaling of the meta-analysis and discussed the depth issue as a major challenge in the model-data comparison.

***Selection of area/climate for simulation is important. By including extra area (where LUC not going to happen) or excluding potential LCC area, you might bias the overall estimation. Authors suggest three different extends and each is not ideal:***

1. ***Entire vegetated area of the land surface (too big. Low chances e.g. for forest on permafrost to be converted to cropland)***

2. ***Area where LUC has taken place historically (too narrow, LUC might came to new places, e.g. tropical deforestation)***

3. ***Where meta-data were available (even more narrow)***

***I would suggest to overlay PFT map with climate one and define climatic patterns where one or another PFT can appear. This would cover all current and potential LUC.***

This comment from the reviewer indicates that our description on how we selected model regions for comparison with the meta-data as described in section 2.4 was not clear. We actually did not suggest that the meta-data can be used to evaluate the entire vegetated land surface as pointed out by the reviewers' approach 1. We have revised this section in the manuscript and added a more detailed explanation on why

we used the other two approaches for selecting the regions for comparison (Approach 2 and 3). The reviewer is correct that these two approaches may not be representative of regions where LUC has not taken place historically. However, the meta-data on LUC exist only in regions where LUC has taken place historically. The only approach not introducing biases is thus to assess regions of LUC, since future LUC may move to regions where the meta-data are not representative. Our study presents a method to identify suitable models that can also be used to project soil carbon changes due to future LUC. However, such projections are beyond the scope of our study, which focusses on model evaluation.

**Minor comments**

***Page 7, line 8. In fact, forest might have lower NPP compare to cropland, but most of the dead matter come to soil surface where decomposition is slow***

We have re-written the sentence to show that we are explaining the reasons for the simulated increase in the model. We have further clarified that the change in soil carbon is driven by on average higher productivity.

***Page7, line 10. Here could be different explanation. Soil respiration is higher in cropland compare to grassland because of the tillage. That is why having similar NPP grassland accumulate more carbon.***

We agree with the reviewer that tillage leads to more soil carbon losses in croplands compared to grasslands. However, the sentence explains the soil carbon response of our standard model simulation, which does not include tillage. We demonstrate the effects of crop management practices with the simulation accounting for crop harvesting

and further discuss the implications in section 4.2.1.

**Page 9, line 10 "Without accounting for crop harvesting" makes sense only to demonstrate how DGVMs are far from reality while not taking into account such evident things like management and disturbances.**

We have rephrased this sentence to indicate that the results are for the sensitivity simulations neglecting burning in our standard model simulation, which does not account for crop harvesting.

**Page 9, line 10 "switching off disturbances in grass... leads to the right direction of soil carbon change" I hope we aim is to describe the reality with the model, but not just have a similar estimation. Disturbances exist. If result is better without disturbances, then the model makes mistake in its different part.**

We agree with the reviewer that the phrasing of this sentence suggests that we switch off burning to get to the right direction of change. We have rephrased the sentence to clarify that this result represents a sensitivity simulation. In addition, we have included an additional sentence to explain that we aim to show that the choice of the vegetation types affected by disturbances in DGVMs has an influence on the soil carbon response to LUC.

**Page 10, Line 1. Grassland and cropland NPP generally larger compare to forest in temperate region also because they allocated on best locations (soil, slope, etc). However if you try to convert existing forest to grassland or cropland you**

**might not get increase of NPP.**

The meta-data include local-scale measurements that are mainly done using paired plots designs; hence such sub-grid scale heterogeinities are accounted for in the meta-data. Therefore, an assessment of the soil carbon response to LUC associated with such heterogeinities is beyond the scope of our study.

---

## Author Comment (AC3) · 24 Aug 2016

[twoside]article

lipsum [sc]mathpazo amssymb, amsmath graphicx [T1]fontenc microtype [final]pdfpages [hmarginratio=1:1,top=32mm,columnsep=20pt,left=26mm]geometry multicol [hang, small,labelfont=bf,up,textfont=it,up]caption subcaption [subfigure]subrefformat=simple,labelformat=simple,listofformat=subsimplebooktabs float hyperref lettrine paralist abstract titlesec

fancyhdr

**Response to T. Pugh of the paper entitled "Soil carbon response to land-use**

[Figure]

change: Evaluation of a global vegetation model using observational meta-analyses"

Ref.: bg-2016-161

Below are the reviewers suggestions (***bold italic font***) and our responses to each point (normal font). In some of our responses, we have cited text from the revised manuscript (*italic font*).

*Nyawira et al. develop a framework to evaluate the response of soil carbon stocks in the DGVM JSBACH to land-use change, using meta-analyses of observations of soil carbon stock change. They find that the baseline model is unable to reproduce the observations for most transition types tested, but that inclusion of crop harvest and exclusion of fires on pasture land notably improves the fit of the model response.*
*The analysis has been carefully executed and the manuscript is well written. It provides a useful framework for evaluation of soil carbon response to land-use change, which is generally poorly evaluated in DGVMs used to provide land-use emission estimates, despite constituting a substantial part of the overall emission. I am happy to recommend publication, subject to addressing the following minor comments.*

Thank you for your comments. We are happy that you find our results worth of publication in Biogeosciences.

*Section 2.2 -In order to understand the differences between the various land-use types considered, more information about the different PFT types is required. In particular, how do C3/C4 grasses, C3/C4 pastures and C3/C4 crops differ from one another? I suggest to add a table listing the differences in any important*

***PFT parameters, or other parameters which may be important to the land-use type (e.g. different soil decomposition rates?).***

We have added two sentences in this section. In paragraph one we explain the difference in terms of photosynthesis: *"The PFTs differ with respect to their phenology, albedo and photosynthetic parameters; photosynthesis is based on Farquhar et al. (1980) for C3 plants and Collatz et al. (1992) for C4 plants."*
In paragraph two we added a sentence to explain that there is no distinction of the decomposition rates among the different PFTs, but only between woody versus green litter. *"Non-woody litter has the same decomposition rates for all the PFTs, while the decomposition of woody litters depends on the woody diameter".*

***Pg. 4, line 21 - I wasn't quite sure here if the forest PFTs had been extended to cover areas not presently covered by forest, or not. Can the authors clarify?***

In the idealized simulations we extended the vegetation type, e.g., forest, to cover the entire globe in our simulations. However, our criteria for the grid cells selections based on climate or historical LUC removes regions where the forest do not exist. We have clarified this in section 2.3.

***Pg. 5, line 12 - 50% seems a high proportion of crop NPP to be allocated below-ground. This will have a substantial influence on the size of the flux lost to harvest and should therefore be discussed in relation to published literature in the discussion section.***

We are sorry this was a mis-communication from our part. There is no 50% above-ground and 50% belowground allocation of NPP in the model. The choice of the 50% harvest is based on root to shoot ratio and not only accounts for root biomass, but also for other unharvestable plant parts. We have also added a discussion on the uncertainties associated with this parameter choice. We have the sentence describing the

aboveground and belowground allocation and added the following sentence;
*The choice to transfer 50% to the litter is approximated from the average root to shoot ratio of several crop types (Extended data Fig. 2 in Gray et al., 2014). The 50% accounts for root biomass, unharvestable parts of the stem biomass being left in the field and a potential return of carbon to soil in the form of manure.*

**Pg. 5, line 27 - I think these are grid-cells where just the relevant transition type has taken place (based on Fig. S2), and not where any LUC has taken place at all? This isn't clear to me from the text. Also, on first reading I thought climate and LUC criterion were being applied simultaneously, and it only later became clear that they were being applied separately.**

We have re-written the sentence to make it clearer that it is where the relevant transition has taken place. In addition, we have made it clear that the two criteria are independent.

**Pg. 6, line 4 - Should beta have units of length? Also, please define the units of $d_0$ (presumably cm).**

We have removed the scaling section from the manuscript.

**Pg. 6, line 13 - How do you sample simulated soil C changes over the ages? Do you take a simple mean over the age range in the observations, or do you weight the mean by the number of observations in each age range? I would argue the second is much better, if you have the data to do it.**

We have added two sentences to clarify that the second approach is indeed taken.
*"For this we use the age represented by each site in the meta-analyses to select the transient years in the simulations to include in averaging the soil C response. We average the soil carbon response over these years and spatially for the selected regions.*

*This average represents the simulated soil C response over the different ages repre-sent in the meta-analyses."*

**Section 3.2, para. 1 - I think you can be a bit more assertive here in saying that the reason for the results from the crop to grass simulations is fire. That seems to be very clearly demonstrated at the end of the paragraph, and I'm not sure why the section beginning "we suspect" (line 11) is included.**

We have re-written the sentence.

**Pg. 8, line 14 - Should Fig. S2 be cited here?**

We have included the figure citation.

**Pg. 9, line 24 - What is meant specifically by "forest floor"? Surface litter?**

Correct. We have changed forest floor to surface litter.

**Pg. 9, line 27 - Is the larger NPP for forests than pastures in accordance with the literature? Would be good to discuss this briefly with some references.**

We added a discussion on this in section 4.2.3.
*"For most of the considered regions in the tropics, the larger simulated NPP for forests compared to pastures is consisted with other observations (Smith et al 2012). Murty et al. (2002) associated the observed increase in soil C following conversion of forest to pasture with low initial content of soil C, application of fertiliser and careful management that avoided overgrazing. Table 3 shows low previous land use soil C for forest to pasture compared to forest to crop in the meta-analyses. However, this is not the case*

*for the simulated soil C in the considered regions."*

***End of section 4.1.4 - Absolutely agree with this sentiment, but shouldn't we then be aiming for a more stringent test than getting within the very large standard deviation that results from this small-scale heterogeneity?***

We agree with the reviewers sentiment. However, in our model-data comparison we do not use the standard deviation as a measure of agreement between the simulated results and the meta-analyses. In this part we discuss that the model may not capture spatial variability in soil C changes due to other missing processes, which the meta-analyses may capture.

***Pg. 11, line 23 - What exactly is meant by "top soil"?***

We have clarified that the top soil refers to the upper 30cm.

***Section 4.3 - I agree with the general statement regarding absolute estimates, but the way this section is written seems to imply that JSBACH was successful in capturing the observations in this evaluation, which I feel would be stretching it a bit for several of the transition types, especially grass to crop (based on Fig. 2)***

The reviewer is right that our phrasing was misleading. We have re-written the section as follows:
*"Even though DGVMs provide land-use-related absolute soil C changes, our comparison focused on relative changes. This is the preferred variable in the meta-analyses because spatial heterogeneity partly cancels in relative terms when two sites in close proximity are compared to each other, as done in chronosequences. Only relative changes allow for deriving robust carbon response functions (Poeplau et al 2011). In the jsb_drvn_harv simulation, the equilibrium changes indicate a decrease in soil C of*

*about 11 kgC m$^{-2}$ and 3 kgC m$^{-2}$ for forest to crop and grass to crop, respectively, in the temperate region. The decrease for forest to crop in the tropics is about 9 kgC m$^{-2}$ (Fig. 1b). The reverse LUCs result in soil C increase of about the same magnitude. Because DGVMs are unaffected by small-scale spatial heterogeneity, their estimates of absolute changes are expected to be more robust than those of meta-analyses and therefore better representative for global carbon responses. After successful evaluation against relative changes, DGVMs can therefore be used to assess large-scale soil C changes in the absolute terms that are relevant for carbon budget estimates."*

***Table 4 - I'm not clear on the logic of having this table in addition to Table 3. It would seem more helpful to add the obs_drvn and jsbach_drvn_harv data to Table 3 (appropriately adjusted for 30 cm depth), to allow them to also easily be assessed against the observations.***

We have removed the model scaling section following concerns raised by another reviewer and included the meta-analyses soil C densities in Table 4 (now Table 3).

---

## Author Comment (AC4) · 24 Aug 2016

[twoside]article

lipsum [sc]mathpazo amssymb, amsmath graphicx [T1]fontenc microtype [final]pdfpages [hmarginratio=1:1,top=32mm,columnsep=20pt,left=26mm]geometry multicol [hang, small,labelfont=bf,up,textfont=it,up]caption subcaption [subfigure]subrefformat=simple,labelformat=simple,listofformat=subsimplebooktabs float hyperref lettrine paralist abstract titlesec

fancyhdr

**Response to E. Marin-Spiotta of the paper entitled "Soil carbon response to land-**

use change: Evaluation of a global vegetation model using observational meta-analyses"

Ref.: bg-2016-161

Below are the reviewers suggestions (***bold italic font***) and our responses to each point (normal font). In some of our responses, we have cited text from the revised manuscript (*italic font*).

Thank you for your comments, which have helped us greatly in improving our manuscript. We would like to clarify that there were some changes requested by one of the reviewers prior to the publication of the manuscript on discussion. Therefore, some of the line numbers in this review refer to the older version of the manuscript at the quick report stage. To be consistent, we have added the new line numbers that match the current manuscript on discussion in parenthesis.

***This study uses published results from temperate and tropical meta-analyses that calculate mean responses of soil carbon change to different land-use transitions from field measurements of paired plots to better constrain estimates of belowground response to land-use change at a global scale simulated by a dynamic global vegetation model. To my knowledge, this is the first time global syntheses of data from published meta-analyses have been used to compare to results from models. The research takes advantage of a large effort to synthesize global temperate and tropical data on soil C to estimate the response of soil C stocks to major land use transitions.***

***Overall, the writing could be improved to more clearly describe the modeling approach and to distinguish it from past efforts. Some sections have minimal***

[Figure]

*text (for example, description of the observational data and the meta-analyses approach), and while the attempt to be concise is appreciated, more information would make it easier for readers to understand and attempt to replicate the approach.*

We have expanded section 2.3 and 2.4 to make it easier for readers interested in replicating the results (see response to B. Guenet). We have also expanded the section describing the meta-analyses (section 2.1) as follows:
*"In this study, we use results from the meta-analyses by Poeplau et al. (2011) in the temperate regions and Don et al. (2011) for the tropical regions including 95 and 385 published studies, respectively. The published studies include sites from different countries in the tropics and temperate regions. The site studies were conducted using two main experimental designs: paired plots comparing soil C between two adjacent sites with different land use types, and time series where the soil C of a particular site was monitored overtime after LUC. The paired plot approach is used to construct chronosequences comprising of plots with different ages after LUC that use one of the plots, with the prior land use, as the reference site. The paired plot based approach goes a long with a higher methodological uncertainity in the data due to differences in the inherent soil properties such as texture between the plots, which affect the response of soil C to LUC. In contrast, the time series observational data are without such uncertainties, but very few time series are available to investigate the response of soil C to LUC. In calculating the soil C changes across the different sites, the reference site was always assumed to be in equilibrium.*
*The meta-analyses defined the following criteria for including the site studies: (1) climate conditions, age of the current land use, and the relevant site characteristics such as soil type, texture and land-use history had to be provided, (2) studies on organic and wetland soils were not included and (3) for paired plots the sites had to be adjacent to each other to reduce uncertainties due to the spatial variability of soil properties unrelated to the LUC (Don et al., 2011; Poeplau et al., 2011). Any studies that did not match*

*any of the criteria were excluded in the compilation. The soil bulk densities were used to calculate the soil organic carbon in Mg/ha. Mass correction was applied to account for changes in density with depth (Ellert and Bettany, 1995). In addition, Poeplau et al. (2011) used different variables, such as climate, time after LUC and the clay content, to derive carbon response functions (CRFs) describing the temporal response of soil C to LUC for the temperate regions. The response functions include general CRFs that account for only time after the LUC and specific CRFs that account for other site properties. Table 1 shows the LUCs represented in the two meta-analyses that are included in our study."*

***The discussion dives directly into details of the model but would benefit from an overall summary highlighting the main findings of the paper and being organized around the take home messages of the research. An effort to frame the discussion in a bigger context will help identify novel insights to a broader audience who may not be familiar with the modeling approach but is still very interested in the findings. For example, consider starting the discussion with section 4.1.4 (which has a great discussion of scale between the models and the field observations) then going into the details of the crop harvest and fire, then discussion of the challenges (current section 4.2.1).***

We have re-organized the discussion section to focuss on three key aspects that are important for the broader audience. (1) The general approach for evaluating DGVMs against the meta-analyses, (2) the causes of model deviation from the meta-analyses that we identified for the DGVM JSBACH, and (3) the challenges that are involved in model-data comparison.

***The use of the term "meta-data" in the title and throughout the paper to represent results from a meta-analyses (a specific statistical test that calculates differences (effect size or response ratio) between data points) is confusing and***

[Figure]

*inaccurate (and distracting) as this term has a different formal definition ("data that describes other data"). To avoid unnecessary confusion, please use an alternate term, such as "field data", "observations", "observational data" or "results from meta-analyses." The term meta-data in the paper is used to refer to meta-analyses (published studies using the specific statistical approach), to the results of these analyses and to general syntheses of published data, further adding to confusion, as these are not the same.*

We have removed the word meta-data and adopted meta-analyses and observational data in our manuscript.

*In a few places, it is unclear how data used in the model simulations is then related to the land cover types and information associated with the observations of soil carbon change. For example, how are the different plant functional types, especially the different forest PFT, related to the 4 idealized land use classes? Given that the observational data used in this paper is heavily biased towards tropical sites (from Don et al. 2011 vs Poeplau and Don 2015), it is expected that the land cover description of the sites in the published literature do not match the PFTs at the global scale in the DGVM.*

This is a good point. The land cover type description is indeed an important factor when comparing soil carbon changes with meta-analyses. In JSBACH there are four forest PFT distinguished in terms of their phenology (broadleaf and decidous) and location (tropical and extratropical). Our grid cell selection criteria ensures that the selected regions include only the PFTs existing in the particular regions. Therefore, the comparison of the simulated soil C changes with the data by Don et al 2011 includes regions with only tropical PFTs, while the comparison with Poeplau et al 2011 includes extratropical PFTs. We added the paragraph below in Section 2.3 to clarify how we derive the distribution of the different PFTs contained in each land cover map.

*"We create idealized land cover maps for four vegetation types; forest, crop, grass and pasture. In these cover maps the entire globe is covered by each of the four vegetation types. The regions where one of these vegetation types does not exist are masked out in our comparison of simulated results to the meta-analyses (see section 2.4). Each land cover map consists of several PFTs: Forest land cover contains evergreen and broadleaf PFTs in the tropical and extratropical regions, while crop, grass and pasture land cover contains both C3 and C4 PFTs. To create the idealized land cover maps we start with a present day JSBACH land cover map obtained by remapping observed vegetation distribution into PFTs (see Friedl et al. (2010) and supplementary material section S1). In the grid cells where two PFTs belonging to the same vegetation type already exist, e.g., in a grid cell with both tropical deciduous and tropical evergreen from observed vegetation distribution, we scale the cover fraction to the entire grid cells based on their relative distribution."*

***In addition, where do the plant productivity measurements used in the model come from and how do these relate to the types of vegetation and their growth rates from the observational soil carbon studies?***

Plant productivity is either simulated directly by JSBACH (standard set of simulations) or prescribed from observations. Section 2.3 describes where these measurements come from (flux net measurements extended globally using machine learning algorithms). See previous response on how the PFT distribution within a land cover type is derived.

***How do the model simulations address uncertainties in field soil C measurements? For example, the values given in section 3.1 are averages with some associated error. What is the size of this error, how does this variability affect the carbon response functions, and how then do these influence modeled re-***

*sults?*

You are correct that field measure can be quite uncertain. However, in our comparison we do not force the model with the observed soil C from the meta-analyses, but just compare the two. Because we do not use observed soil C as forcing in our model, there are no propagated errors from the meta-analyses to our simulated results. The carbon response functions used are derived from the meta-analyses. The standard deviation provided in our comparison provides a measure of the spread in the considered regions and sites. Therefore, assessing the error associated with the meta-analyses is outside the scope of our study.

We have added a sentence to discuss uncertainties associated with the methodological designs used to obtain the observational-data in the meta-data analyses (section 2.1).

***Page 1, line 23: Soil C changes with LUC are not only influenced by differences in inputs, but also outputs, and alteration to processes that store C in soils.***

We have re-written the sentence as follows;
*"Soil C changes due to LUC are caused by changes in soil C inputs and outputs when one vegetation type is replaced by another. Changes in soil C inputs stem from differences in litter quality and quantity, while the changes in outputs stem from alteration of soil decomposition processes that govern stabilisation of carbon in soils."*

***The last paragraph of section 4.1.1 in the discussion briefly starts to address other factors that can influence soil C decomposition that are not included in the model. Some further discussion on how the focus on plant litter chemistry and climatic variables as controls on C cycling that is the basis for the biogeochemical component of the model and the absence of other mechanistic controls on soil C turnover and how they may influence differences between simulation***

*results and observational data would enhance the paper.*

We have extended the discussion in section 4.1.1.
*"The carbon model used in this study simulates soil C based on the plant chemistry and climate. Recent studies have shown that the inclusion of microbial dynamics and priming processes in biogeochemical models can improve model agreement with observations (e.g., Wieder et al., 2013). As these processes are different across land-use types, the inclusion of such processes in future generation of DGVMs may lead to improved simulated soil C response to LUC."*

*Page 4, line 13-20 (line 2-10): This paragraph discusses grid cells with only one vegetation type and also proportions of grid cells undergoing different land use transitions from different vegetation types. Please clarify which approach was taken in the paper and distinguish between old approaches (for example, additive soil C pools with LUC) and the new one proposed in this study.*

We have re-written the paragraph to clarify that we use idealized and not realistic LUC simulations. The goal of this paragraph is also to discuss why we need to do idealized simulations and not use realistic LUC simulations in evaluating DGVMs.
*"We perform idealized LUCs in which only one vegetation type covers the entire globe and which is subsequently to another type. The idealized simulations approach prevents interference of soil C changes that occur due to different types of LUCs occuring simultaneously in a grid cell or due to sequences of LUC over time. Such interferences occur in realistic LUC simulations. Here, most grid cells in the globe contain a mixture of different vegetation types and at a given year different LUCs may occur. For example, part of the forest in a grid cell may be converted to crop and at the same time part of the grass be converted to crop. Many DGVMs do not separate the soil C for the different PFTs and have one soil C pool for all the PFTs. Those that separate the soil C, e.g. JSBACH, typically add the soil C of the old PFT to the new PFT after LUC.*

*Therefore, soil C change resulting from a specific LUC cannot be obtained using such realistic simulations. The idealized simulations approach used in this study ensures that starting with equilibrium soil C from one land use then changing to another land use, the resulting soil C change can be associated with the specific LUC."*

***Page 5, line 3 (Page 4, line 26): What are the four idealized land use cases? These could be identified here or earlier in the description of the observational data and the meta-analyses.***

We have clarified that the four idealized land use cases refer to crop, forest, grass and pasture.

**Minor comments**

***Page 1, line 5-11: Consider the following sentence reorganization: "Our simulated results show model agreement with the observational data on the direction of changes in soil carbon for some land-use changes, although the models generally estimated smaller magnitudes of change. The conversion of crop to forest resulted in a simulated soil carbon gain of 10% compared to a gain of 42% in the data, whereas the forest to crop change resulted in a simulated loss of -15% compared to -40%. The model and field data disagreed for the conversion of crop to grassland. The simulations estimated a small soil carbon loss (-4%), while field data indicate a 38% gain in soil C with the same land-use transition. These model deviations from the observations are substantially reduced by explicitly accounting for crop harvesting and removing burning in grasslands from the model."***

We have adopted the suggested changes and re-written the section as follows;
*"Our simulated results show model agreement with the observational data on the di-*

*rection of changes in soil carbon for some land-use changes, although the model simulated generally smaller magnitude of changes. The conversion of crop to forest resulted in soil carbon gain of 10% compared to a gain of 42% in the data, whereas the forest to crop resulted in a simulated loss of -15% compared to -40%. The model and the observational data disagreed for the conversion of crop to grasslands. The model estimated a small soil carbon loss (-4%), while observational data indicate a 38% gain in soil C for the same land-use change. These model deviations from the observations are substantially reduced by explicitly accounting for crop harvesting and neglecting burning in grasslands in the model."*

***Page 1, line 17: suggest deleting: "(hereafter meta-data)" as this is an incorrect use of this term.***

We have deleted this and adopted meta-analyses throughout the manuscript.

***Page 1, line 18: add references to the meta-analyses here***

We have added an example reference for the meta-analyses here. The full list of references is provided later in paragraph 3 in the introduction where we discuss the meta-analyses.

***Page 2, line 7: Rewrite: "Despite the dependence of the soil carbon response to local conditions of soils, climate and management practices, regional and global syntheses of published data can be useful to aggregate local-scale measurements on soil carbon changes and estimate mean responses to different***

*LUCs using a meta-analyses approach."*

We have re-writen the sentence as suggested.

*Page 2, line 8 and 10: Here meta-data should be replaced by meta-analyses.*

See our response to the terminology concern.

*Page 2, line 10 and Page 3, line 7: what is meant by "harmonize"a temperate ? Can you use another term?*

We have re-written the sentences and removed the term "harmonize".

*Page 2, line 15: Marin-Spiotta and Sharma (2013)'s work did not use a meta-analyses approach*

We have removed the reference from the paragraph describing meta-analyses.

*Page 3, line 5 (line 3): replace "the meta-data" with "results from the meta-analyses"*

We have replaced meta-data with results from the meta-analyses.

*Page 3, line 6 (line 4): The "quality criteria" sentence structure is awkward. Consider: "These meta-analyses were conducted on paired plots of similar soil type and texture, to reduce uncertainties from hetereogeneous soil properties unre-*

*lated to the land-use transition."*

See the response to the meta-analyses section (section 2.1).

*Page 3, line 30 (line 21): Replace windbreak (check definition) with windstorm.*

We have replaced windbreak with windthrow.

*Page 4, line 3-5 (Page 3, line 22-25): Consider rewording: "The decomposition rate of litter is controlled by its chemical composition, as determined by its solubility (acid, water, ethanol and non-soluble hydrolysable pools) and the presence of a slow decomposing humus pool." It is unclear from the text whether the humus pool is part of the plant litter or a soil organic matter pool? Does the model include above and belowground litter pools?*

We have clarified that all the litter pools and the humus pools in YASSO are treated as part of soil organic matter. We have added two sentences to explain the difference between above and belowground litter.
*"Additionally, litter is split into aboveground and belowground where the aboveground litter burns while belowground litter does not. All the litter pools–aboveground and belowground–and the humus pool are summed up in obtaining the total soil carbon."*

*Page 4, line 26 (line 15): replace "ran" with "run"*

We have replaced ran with run in the sentence.

*Page 10, line 6: above refers to what?*

We have removed the term above in the restructuring of the discussion section.

*Figures 1 and 2 are hard to read. Consider that the green, orange and brown col-*

*ors will be difficult to distinguish for readers with color-blindness, which affects almost 10% of males in many European and English-speaking countries. The grey background also reduces the contrast between the lines, and the lines are too small and hard to see.*
*Figure 3. See earlier comment about choice of colors.*

We have changed the colours in the figures to colours that aid in color-blindness. Additionally we have increased the thickness of the different lines.